# Polymer architecture dictates multiple relaxation processes in soft networks with two orthogonal dynamic bonds

Sirui Ge[1,2], Yu-Hsuan Tsao[1,2] & Christopher M. Evans ⬡ [1,2,3] ✉

Materials with tunable modulus, viscosity, and complex viscoelastic spectra are crucial in applications such as self-healing, additive manufacturing, and energy damping. It is still challenging to predictively design polymer networks with hierarchical relaxation processes, as many competing factors affect dynamics. Here, networks with both pendant and telechelic architecture are synthesized with mixed orthogonal dynamic bonds to understand how the network connectivity and bond exchange mechanisms govern the overall relaxation spectrum. A hydrogen-bonding group and a vitrimeric dynamic crosslinker are combined into the same network, and multimodal relaxation is observed in both pendant and telechelic networks. This is in stark contrast to similar networks where two dynamic bonds share the same exchange mechanism. With the incorporation of orthogonal dynamic bonds, the mixed network also demonstrates excellent damping and improved mechanical properties. In addition, two relaxation processes arise when only hydrogen-bond exchange is present, and both modes are retained in the mixed dynamic networks. This work provides molecular insights for the predictive design of hierarchical dynamics in soft materials.

Viscoelasticity is one of the most important properties of polymers which arises from the complex structure and macromolecular nature of the chains[1]. To understand viscoelastic properties of polymers, dynamic mechanical testing methods such as rheological measurements, i.e., small amplitude oscillatory shear, are commonly used to understand the relative solid-like ($G'$, storage modulus) and liquid-like ($G''$, loss modulus) contributions to the overall frequency dependent response[2]. One of the key features of viscoelastic spectra is the presence of peaks in the loss modulus due to different relaxation processes of polymers such as segmental relaxation and terminal or chain relaxation[3]. The loss peaks also correspond to energy dissipation and sound/vibration in damping materials when they occur at a similar frequency to the input wave[4]. However, it is still a challenge to predictively design damping materials with tunable loss spectra (i.e., breadth, shape, number of relaxation modes) because of the many competing factors such as glassy dynamics, chain relaxation, polymer chemistry, architecture, and presence of other dynamic processes such as bond exchange or dissociation. Connecting molecular structure to macroscopic viscoelasticity would enable improved design of polymers for tissue engineering, energy dissipation, adhesion, additive manufacturing, and novel mechanical properties. Introducing multiple dynamic bonds to control key aspects of viscoelasticity is a promising route towards this goal.

Many variations of dynamic covalent bonds have been incorporated into networks, such as boronic esters[5], imines[6], and vinylogous urethanes[7], as well as various non-covalent interactions such as hydrogen bonding[8], metal-ligand coordination[9], and ionic bonds[10]. The introduction of dynamic bonds provides materials with unique properties such as self-healing[11], super-stretchability[12,13], adhesive properties[14] and toughness mainly due to the significant modification

[1]Department of Materials Science and Engineering, University of Illinois Urbana Champaign, Champaign, IL, USA. [2]Materials Research Laboratory, University of Illinois Urbana Champaign, Champaign, IL, USA. [3]Beckman Institute for Advanced Science and Technology, University of Illinois Urbana Champaign, Champaign, IL, USA. ✉e-mail: cme365@illinois.edu

of the viscoelastic properties. Usually, the dynamic bonds prolong the terminal relaxation time due to the formation of a transient network, and it is the bond exchange process that ultimately determines the overall timescale of network rearrangement[15,16]. The incorporation of kinetically distinct dynamic bonds is a powerful route to imparting additional relaxation processes at distinct timescales, resulting in hierarchical relaxation modes and the ability to design complex viscoelasticity in soft materials[17]. The polymer architecture also plays an important role in determining the overall properties of dynamic networks[18]. Sumerlin and coworker found that the incorporation of permanent crosslinker in a dynamic network can maintain the structural integrity but still possessed exceptional self-healing and reprocessibility[19]. Also, they discovered that a block copolymer architecture reduces the macroscopic flow at large deformation when bonds are localized in one domain[20]. Konkolewicz et al. reported that crosslink distribution within a block polymer tends to demonstrate phase separation and displays better creep resistance[21]. Besides mechanical properties, the overall viscoelastic behavior has been reported to be correlated with the polymer architecture on dynamic polymers with urea-urethane hydrogen bonds[22].

Using multiple dynamic bonds to control macroscopic viscoelasticity has been pursued in a limited number of prior studies. The boronic ester bond is a model chemistry because bond exchange dynamics can be readily tuned by changing the substituents, neighboring groups, or pH in a hydrogel[16]. In one key example using hydrogels, a single network of 4-arm tetra-polyethylene glycol linkers was investigated with kinetically distinct fast and slow boronic esters. Regardless of the ratio of fast/slow bonds, only a single relaxation mode was exhibited by the gel[23]. In a separate study, two kinetically distinct boronic esters were incorporated into solvent-free telechelic polydimethylsiloxane (PDMS)[24] and only one relaxation mode was observed. Thus, mixing fast and slow bonds does not guarantee well-separated relaxation modes in polymer networks. Conversely, multiple relaxation modes have been achieved in 4-arm tetra-polyethylene glycol hydrogels with two kinetically different metal-ligand crosslinkers where two relaxation peaks appear which systematically varied with the concentration of each bond[17]. When mixed Meldrum acid-derived crosslinkers with distinct dynamics were introduced into pendant PDMS networks, multiple relaxation modes were exhibited in one case where the crosslinkers had very different kinetics, but not in other cases[25]. The roles of bond exchange mechanisms and network architecture on the resultant viscoelasticity of networks with multiple dynamic bonds is not well understood.

The incorporation of two orthogonal dynamic bonds, which participate in distinct and independent bond exchange reactions, into one polymer network is one strategy to generate multiple and well-separated relaxation modes. Here, the term orthogonal means that one dynamic bond type does not cause exchange of the other in the system. This does not mean that the dynamics are necessarily uncoupled when multiple bonds are on the same chain. Hydrogen bonding is a popular non-covalent interaction that forms binary association through two complimentary motifs[26], and the bond strength can be tuned by increasing the number of H-bonding centers attached to the molecular constituents[27]. 2-ureido-4[1H]-pyrimidone (UPy) is a quadruple hydrogen-bonding moiety which demonstrates relatively high bond strength[28] and a slower exchange rate compared with hydrogen bonds formed through other motifs such as ureas and carboxylic acids[29]. This bond exchange is still relatively fast compared to many dynamic covalent bonds, and a large separation of timescales may be achieved in mixed systems. Such an orthogonal incorporation strategy has been pursued in PnBA networks with pendant UPy stickers and dynamic covalent boronic ester crosslinkers[30]. Only one relaxation mode showed up, potentially due to the close proximity of exchange rates of the two dynamic groups and possible uneven distribution of the functional group along the backbone. Besides boronic esters, the

imine bond is another popular dynamic covalent bond formed through the reaction of aldehyde and amine groups[31]. The bond exchange can occur through metathesis of two imine groups or reaction with free amines[32]. The orthogonal incorporation of imine crosslinkers and hydrogen-bonding motifs was examined by Lehn and coworkers with linear PDMS containing both dynamic bonds and they found that the product demonstrated self-healing properties[33]. Such orthogonal incorporation was also conducted in telechelic PDMS and polypropylene glycol backbones as well. The resulting mixed networks showed excellent self-healing and tailorable properties which can be used for generating wearable sensors[34]. Guan and coworkers improved the mechanical properties of their self-repairable dynamic polymer by introducing hydrogen bonds as a sacrificial bond[35]. Konkolewicz and coworkers incorporated three different dynamic bonds (two different dynamic covalent bonds and a hydrogen bond) into the polymer backbone and studied their mechanical properties[36,37]. However, hierarchical dynamics were not studied in these works. A recent review article summarizes the state of networks with multiple dynamic bonds[38].

Here, two orthogonal dynamic bonds, the quadruple hydrogen bond formed from UPy motif and dynamic covalent imine bonds, are incorporated into both telechelic and pendant PDMS to demonstrate the role of mixed exchange mechanisms and network architecture on viscoelasticity. Two well-separated relaxation modes are exhibited over a broad range of ratios of these dynamic bonds even in telechelic networks, in contrast to prior studies where fast and slow bonds share an exchange mechanism[19]. The fast relaxation mode (from exchange of UPy) and slow mode (from imine exchange) are clearly resolved by oscillatory shear rheology and the corresponding relaxation spectrum. We also show that UPy bond exchange alone can result in two adjacent relaxation modes in mixed PDMS networks, depending on the architecture of the network, which is a key new insight on the viscoelastic design of soft materials.

## Results

### Formation of dynamic networks

Commercially available PDMS from Gelest Inc., with either pendant amines or telechelic end functionalized amines, was chosen as the backbone. For pendant PDMS, two backbones with different chain length and different molar percentage of amine groups (Product Code: AMS-152 and AMS-162) were utilized. For telechelic PDMS, one backbone (Product Code: DMS-A21) was utilized and by definition has two amine groups per chain. The molar percentage of amine was determined from $^1H$ NMR. The molecular weight and polydispersity (PDI) of both pendant and telechelic precursor were determined from gel permeation chromatography (GPC). Since the amine group connecting to the backbone can result in the adsorption of polymers on the GPC column, which prolongs the elution time and lower the molecular weight[39], both pendant and telechelic backbone was treated by BOC anhydride before the GPC characterization. The detailed synthesis method is described in Section 1 of Supplementary Information with synthesis route (Supplementary Fig. 1a) and NMR result (Supplementary Fig. 1b). The GPC traces for all PDMS precursors are demonstrated in Supplementary Fig. 1c. The peak at the short elution time is the sample peak from which the molecular weight and PDI of the PDMS precursors was estimated. The sharp peak at the long elution time is the solvent peak (chloroform). Based on the measured number average molecular weight and the molar percentage of amine groups, the number of amine groups per chain is nearly identical. The sample information about the PDMS precursors is shown in Table 1.

First, pendant and telechelic PDMS with a single dynamic bond type were synthesized. The amine groups on PDMS precursors were reacted with either 2-(1-Imidazolylcarbonylamino)−6-methyl-4-[1H]-pyrimidinone (UPy-CDI) to form UPy or reacted with an aromatic dialdehyde to form imine networks. Then, mixed dynamic networks

were synthesized as shown in Fig. 1. The UPy was introduced to the PDMS backbone first followed by the introduction of imine bonds. The detailed synthesis routes are described in the Method part. Pendant and telechelic PDMS with different imine/UPy ratios are named Pend-k-m/n in which k denotes the molar percentage of the amine on the pendant PDMS precursor, and m,n denotes the percentage of amine groups functionalized with imine and UPy motifs, respectively. Telechelic networks are named Tele-m/n in which m,n denotes the percentage of amine groups functionalized with imine and UPy groups, respectively.

### Glass transition temperatures of dynamic polymer networks
The glass transition temperature ($T_g$) was measured by differential scanning calorimetry (DSC) to control for any variations that occur upon crosslinking. The heat flow curves for pendant (Fig. 2a, b) and telechelic PDMS (Fig. 2c) with varying ratios of mixed bonds (or pure networks) all demonstrate an endothermic step at ~150 K corresponding to $T_g$. Increased crosslink density raises $T_g$, but it does not discernably vary with the different compositions of dynamic bonds (Supplementary Table 1). In addition, the increase of the $T_g$ is linearly proportional to the initial $NH_2$ molar percentage of the polymer backbone (Fig. 2d). This agrees with a statistical theory that the increasing of the $T_g$ in polymers with dynamic bonds is only correlated with the molar fraction of the functional groups, regardless of telechelic or pendant architecture[40]. The theory was also verified by experiments recently[41]. For telechelic PDMS, the melting peaks (Fig. 2c) indicate the crystallization of the PDMS backbone which is far below the temperature used for rheology measurements.

### Absence of Visible Microphase Separation
Microphase separation has been reported in polymers with the UPy motif[42], attributed to the stacking of groups caused by an extra urea introduced from reacting a primary amine with an isocyanate functionalized isocytosine building block[43]. By using the carbo-imidazole

activated methyl isocytosine, this extra urea can be avoided[44]. The absence of microphase separation in the present networks was verified by small angle X-ray scattering (SAXS) measurement (Supplementary Fig. 2a). If microphase separation occurred, a scattering peak would show up at q ~ 1 nm⁻¹ indicating the distance between the microphase separated clusters[45]. As a counterpart, we also synthesized telechelic PDMS with an extra urea group between the PDMS backbone and UPy motif (detailed synthesis method is described in Section 2 of Supplementary Information with synthesis route (Supplementary Fig. 2d) and NMR result (Supplementary Fig. 2e). A prominent SAXS peak shows up, indicating microphase separation, in contrast to mixed PDMS (Pend-3.5-25/75) or pure PDMS (Pend-3.5-0/100) (Supplementary Fig. 2a) in the present study. Thus, the mixed samples in this work have no detectable microphase separation.

### Two distinct relaxation modes from linear viscoelasticity measurement
The complex modulus was measured from oscillatory shear rheological frequency sweeps, beginning with the pendant PDMS networks (Fig. 3a) shown in Fig. 3b and c. In the pure networks with only the UPy or imine, a large separation of the terminal relaxation timescale (determined from the crossover of $G'$ and $G''$) of ~ 4 orders of magnitude is observed. This is expected as imine bonds and UPy have larger differences in bond exchange rate compared to the prior work in which boronic acids and UPy were involved[46]. When mixed dynamic bonds are incorporated into pendant PDMS, the rheological spectra unambiguously demonstrate two distinct relaxation modes as two peaks in $G''(\omega)$ and two rubbery plateaus in $G'(\omega)$ (Fig. 3b, c). The fast one is from the maximum in the $G''$ curve at higher frequency, and the slow one is the crossover time at lower frequency. These two relaxation modes are attributed to the fast UPy bond exchange ($\tau_{fast}$) and the slow imine bond exchange ($\tau_{slow}$), respectively. As shown in Fig. 3b and c, the fast relaxation mode (with timescale of $\tau_{fast}$) corresponds to the terminal relaxation timescale of a pure UPy network and the slow relaxation mode corresponds to the terminal relaxation timescale of imine bond exchange.

To verify that the fast relaxation mode stems from UPy bond exchange, we also synthesized pendant PDMS network with only 50% of amine group reacted with imine crosslinker and 50% free amines (no UPy). The shear modulus spectra demonstrates only one peak corresponding to imine bond exchange (Supplementary Fig. 3). With free amine groups, the imine exchange becomes faster, as the exchange mechanism includes imine transesterification in addition to metathesis[32]. Meng et al. synthesized a pendant PDMS network functionalized with UPy and a permanent crosslinker[47]. In the shear modulus spectra measurement, a relaxation peak in $G''$ spectra indicating UPy bond exchange can also be observed at a similar frequency as the observed $G''$ high frequency peak in our mixed network sample Pend-

**Table 1 | Architecture, Product the Code, number average molecular weight ($M_n$), Polydispersity Index (PDI), Molar percentage of amine group, and Number of amine groups of all the PDMS precursor utilized in this research**

| Architecture | Product Code | $M_n$ (Da)[a] | PDI[a] | $NH_2$ mol%[b] | $NH_2$ groups per chain |
|---|---|---|---|---|---|
| Pendant | ASM-152 | 7332 | 2.08 | 3.5% | 3.4 |
|  | AMS-162 | 5000 | 1.89 | 5.8% | 3.8 |
| Telechelic | DMS-A21 | 4992 | 1.95 | 2.9% | 2 |

[a]Determined by GPC.
[b]Determined by ¹H NMR.

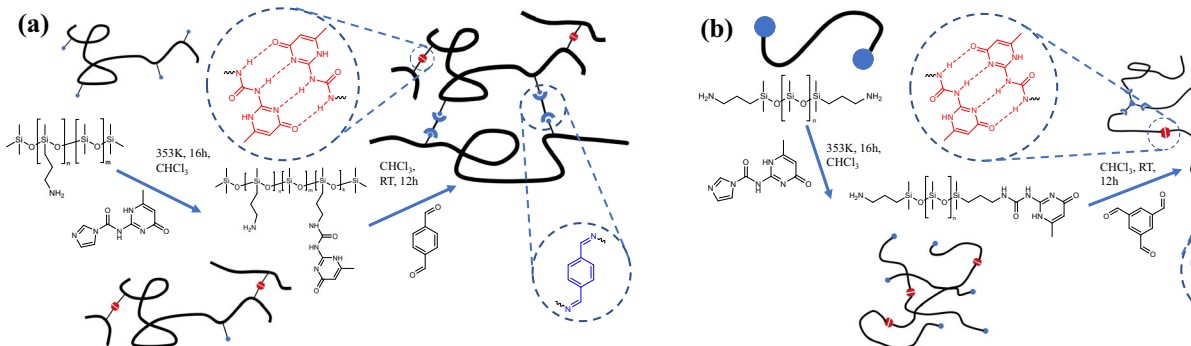

**Fig. 1 | Synthesis route of PDMS with mixed dynamic bond. a** Synthesis route of pendant PDMS with mixed dynamic bond. **b** Synthesis route of telechelic PDMS with mixed dynamic bond.

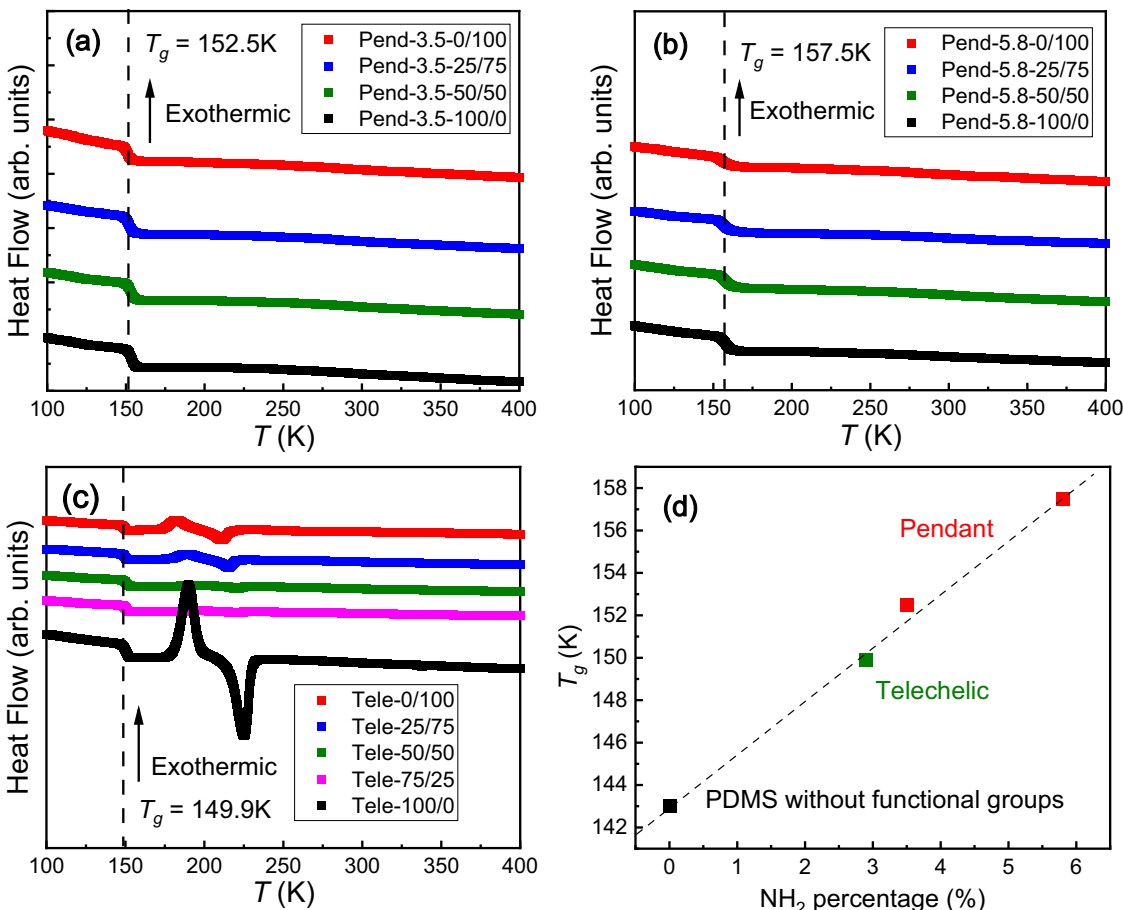

**Fig. 2 | DSC measurement results for all PDMS networks. a** Heat flow curves for Pend-3.5 with different dynamic bond ratios (different colors). **b** Heat flow curves for Pend-5.8 with different dynamic bond ratio (different colors). **c** Heat flow curves for telechelic PDMS with different dynamic bond ratios (different colors). The step on each curve is labeled by dashed line indicating $T_g$ for each polymer. The value of $T_g$ is labeled in each graph. The exothermic direction of heat flow is labeled by an arrow in each graph. **d** $T_g$ as a functional of initial $NH_2$ percentage on the PDMS backbone (Pendant: red; Telechelic: green; PDMS without functional group: black). The $T_g$ of PDMS without any functional groups corresponds to PDMS with molecular weight of 5940 g/mol[70].

5.8-50/50 (Supplementary Fig. 4) and indicates the presence of a permanent crosslinker does not significantly impact the timescale. The prominent multiple relaxation modes in mixed dynamic bond pendant PDMS networks is attributed to the orthogonal exchange mechanisms of UPy and imine (unlike the hydrogels with a single mode) and large contrast in bond exchange kinetics. This work also agrees with one of the samples of pendant PDMS with Meldrum's acid, where multiple relaxation modes exhibits when there is a separation of individual relaxation rates by several orders of magnitude[25].

In terms of the rubbery plateau modulus, the high frequency plateau is independent of the ratio of dynamic bonds. It is only dependent on the initial $NH_2$ molar percentage which sets the crosslink density (Fig. 3d), indicating that the network at this stage is formed through both hydrogen bonds and imine bonds and on the timescale of the experiment, the bonds are effectively static. The low frequency plateau is determined by both initial $NH_2$ percentage and imine bond percentage (Fig. 3d), indicating that the network structure before the terminal relaxation is determined by crosslinked density of imine network and the imine bond exchange results in the final terminal relaxation process.

The shear modulus spectra for pendant samples were measured as a function of temperature to monitor the shift in relaxation times. The fast relaxation mode ($\tau_{fast}$) comes from UPy bond exchange in the mixed sample and is determined from the higher frequency peak position in the $G''(\omega)$ spectra. The timescale of the slow relaxation mode ($\tau_{slow}$) in the mixed network, or the only mode in the pure networks, is

determined from the crossover of $G'$ and $G''$. For pendant PDMS, the fast relaxation process in the mixed sample agrees with the terminal relaxation process in its pure UPy counterpart throughout the measured temperature range with only small deviations (Fig. 3e), indicating that the UPy bond exchange in the mixed pendant network is not affected by the presence of imine crosslinkers. The slow relaxation process in the mixed network becomes slightly faster in the pendant network with imine/UPy ratio of 25/75 compared with the terminal relaxation in the pure imine counterpart, but with less than one order of magnitude difference. With lower imine percentage, networks form with lower effective crosslink density, which results in not only lower plateau modulus but also a shorter imine bond exchange timescale.

Next, the role of backbone architecture was investigated using telechelic PDMS (Fig. 4a) to determine if two relaxation modes can be observed when orthogonal bonds are used. When the molar percentage of imine is higher than 50%, the mixed telechelic network exhibits two well-separated relaxation modes from both UPy and imine bond exchange (Fig. 4b). This is in stark contrast to our prior work on telechelic PDMS networks where two boronic esters were used, indicating the importance of shared versus mixed mechanisms[24]. The high frequency plateau in the present telechelic networks (Fig. 4b) demonstrates lower mechanical modulus than that of both pendant PDMS samples due to lower concentration of overall functionalized groups on the telechelic structure. The low frequency plateau modulus can also be tuned by the ratio of dynamic bonds. If the molar percentage of imine is lower than 50%, however, only the fast relaxation mode from

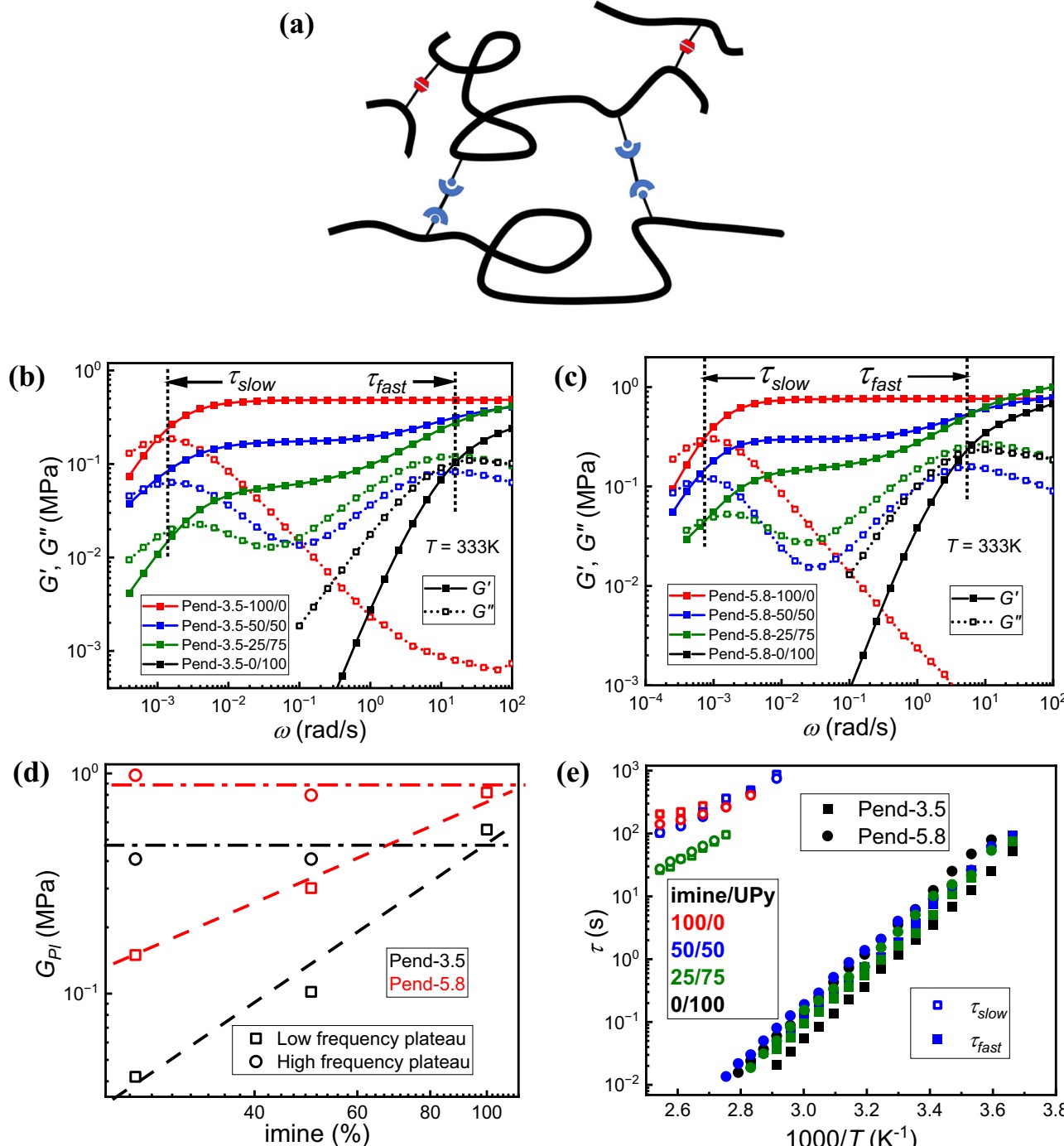

**Fig. 3 | Shear modulus result of pendant PDMS networks. a** Chemical structure of the pendant PDMS networks. **b** $G'(\omega)$ (closed symbol with solid line), $G''(\omega)$ (open symbol with dotted line) spectra of Pend-3.5 with different bond ratios (different colors) measured at 333 K. The position of $\tau_{fast}$ and $\tau_{slow}$ are labeled by dotted lines and arrows. **c** $G'(\omega)$ (closed symbol with solid line), $G''(\omega)$ (open symbol with dotted line) spectra of Pend-5.8 with different bond ratios (different colors) measured at 333 K. The position of $\tau_{fast}$ and $\tau_{slow}$ are labeled by dotted lines and arrows.

**d** Comparison of low frequency rubbery plateau modulus (open square) and high frequency rubbery plateau modulus with different dynamic bond ratios (imine%) and different initial molar percentage of $NH_2$ (Pend-3.5: black, Pend-5.8: red). **e** Temperature dependence of $\tau_{fast}$ (closed symbols) and $\tau_{slow}$ (open symbols) for samples with different precursors (Pend-3.5: square, Pend-5.8: circle), and different imine/UPy ratio (different colors).

UPy bond exchange shows up in the telechelic system (Fig. 4b). Indeed, whether the slow relaxation mode from imine bond exchange shows up is determined by whether the network formed by imine bonds can percolate throughout the material. In the case of pendant PDMS, the imine network structure can be reached with any amount of imine crosslinker as one sticker per chain is enough to form a gelled network with pendant architectures[48]. For telechelic systems, the Flory-

Stockmayer theory[49,50] can be used to determine when percolation of the imine network occurs. The percentage of $NH_2$ reacting with the trialdehyde imine crosslinker in telechelic PDMS needs to reach 50% (for details, see Section 15 of Supplementary Information). Below this threshold, only hyper branched structures with imine crosslinkers as junctions can form so that UPy bond exchange is enough to result in macroscopic flow, as seen in the viscoelastic behavior of Tele-25/75

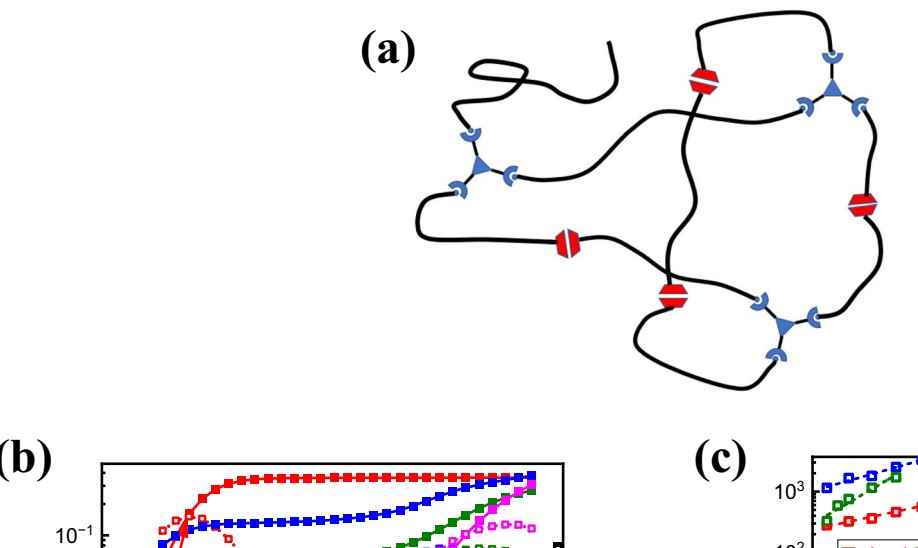

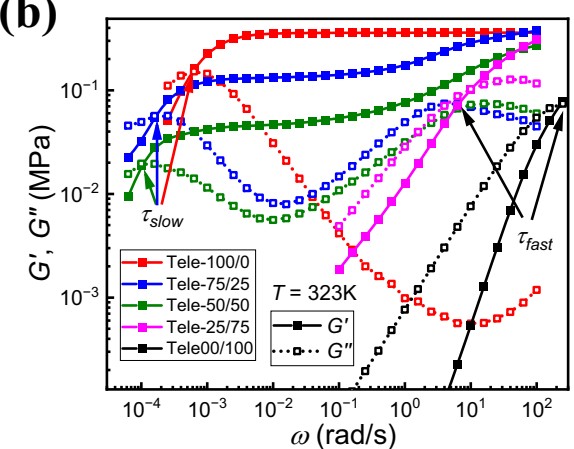

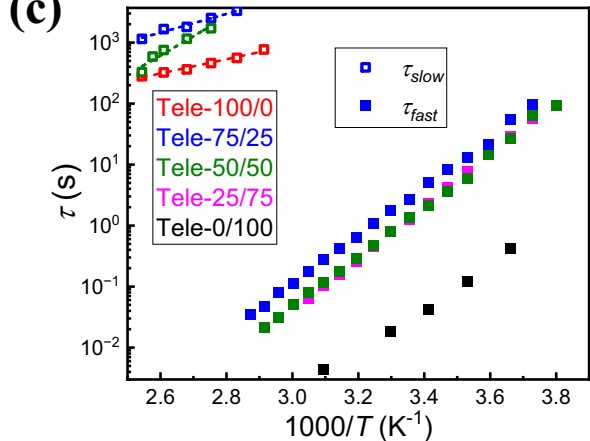

Fig. 4 | Shear modulus result of telechelic PDMS networks. a Chemical structure of telechelic PDMS networks. b $G'(\omega)$ (close symbol with solid line), $G''(\omega)$ (open symbol with dotted line) spectra of telechelic PDMS with different component (different colors) measured at 323 K. The position of $\tau_{fast}$ and $\tau_{slow}$ are labeled by arrows. c Temperature dependence of $\tau_{fast}$ (closed symbols) and $\tau_{slow}$ (open symbols) for telechelic network with different bond ratios (different colors). The dashed line indicates fits using the Arrhenius equation (Eq. 1).

which demonstrates only one fast relaxation mode. Interestingly, the timescale of both fast and slow processes in the mixed network increases compared with those in their pure counterparts. The introduction of imine crosslinkers changes the overall network structure, which prolongs the UPy bond exchange timescale ($\tau_{fast}$). On the other hand, it is different from the case of the pendant mixed network where the terminal relaxation time ($\tau_{slow}$) also increases in the mixed telechelic network. This is attributed to the telechelic architecture as well as the usage of the 3-arm crosslinker. Amin et al. proposed a partner exchange mechanism to illustrate the bond exchange process for telechelic polymers in the case that more than two chain ends "aggregate" together. The model considers that two "aggregates" merge into a bigger cluster and then separate into two new ones, during which the chain-ends exchange their partners[51]. In our case, the chain-ends (amine group) aggregate on the 3-arm crosslinker, which agrees with the case of Amin et al. With fewer 3-arm crosslinkers per volume, it becomes more difficult for two crosslinkers to encounter, leading to an overall increase of terminal relaxation times ($\tau_{slow}$).

The shear modulus spectra were also measured at various temperatures, and the terminal relaxation time ($\tau_{slow}$) for the mixed telechelic network, especially for the Tele-50/50 sample, demonstrates a stronger temperature dependence compared with that for the pure telechelic imine network Tele-100/0. The Arrhenius equation:

$$\tau(T) = \tau_0 \exp\left(\frac{E_a}{RT}\right) \tag{1}$$

where $E_a$ is an activation energy, $\tau_0$ is a prefactor, and $R$ is the gas constant was employed to analyze the activation anergy of both imine bond exchange based on the terminal relaxation time ($\tau_{slow}$) and UPy bond exchange based on the fast relaxation mode at high frequency ($\tau_{fast}$) (Fig. 4c), the result of which is demonstrated in Table 2 with error bars.

For telechelic PDMS with only imine crosslinker (Tele-0/100), the activation energy is relatively small compared with that for UPy bond exchange, which is close to the activation energy reported in the literature about various imine networks[52,53]. With lower density of the 3-arm crosslinker, the activation energy for imine bond exchange becomes higher, and a similar result has been reported by Liu et al. which explains that lower bond density results in the lower chance of

**Table 2 | Activation energy of imine bond exchange (slow mode) and UPy bond exchange (fast mode) for each telechelic PDMS sample**

| Mode | Sample | $E_a$ (kJ/mol) |
|------|--------|----------------|
| Slow | Tele-100/0 | 23.9 ± 0.9 |
| | Tele-75-25 | 29.4 ± 2.6 |
| | Tele-50/50 | 60.7 ± 8.8 |
| Fast | Tele-75-25 | 76.1 ± 0.6 |
| | Tele-50/50 | 79.1 ± 0.8 |
| | Tele-25/75 | 83.1 ± 0.3 |
| | Tele-0/100 | 66.6 ± 2.8 |

collision of two different imine groups[54]. This also agrees with the partner exchange mechanism proposed by Amin et al.[51]. For the fast relaxation mode (from UPy bond exchange), (i.e. $\tau_{fast}$), all mixed telechelic networks demonstrate a similar timescale and activation energy with slightly slower dynamics for telechelic PDMS (75/25) (Fig. 4c). This timescale is more than one order of magnitude slower than $\tau_{fast}$ of the pure UPy network (Tele-0/100 sample) (Fig. 4c), which is consistent with the shear modulus spectra in Fig. 4b. In addition, the activation energy for UPy bond exchange in the mixed samples is slightly higher. This also clearly demonstrates how the UPy bond exchange process in mixed telechelic networks is retarded by the introduction of imine crosslinkers.

### Improved mechanical and damping properties in the mixed network

The mixed networks with both pendant and telechelic architecture display well-resolved multiple relaxation modes, which can lead to excellent damping properties and improved mechanical properties. The damping properties of materials is evaluated through $\tan\delta = \frac{G'}{G''}$. Typically, $\tan\delta$ greater than 0.3 is considered a criterion for sufficient damping properties of polymeric materials[55]. Comparing with the network with only imine bond which only displays the turn-up of $\tan\delta$ at low frequency corresponds to imine bond exchange, the mixed network, however, demonstrates additional $\tan\delta$ peak at low frequency corresponds to UPy bond exchange (Fig. 5a–c), indicating the improvement of the damping properties. The strength of the peak increases and reaches, or exceeds, 0.3 when more than 50% of the UPy bonds are present. It is difficult to quantify the magnitude of the imine damping peak, because it coincides with flow which leads to an abrupt increase in $\tan\delta$. Another metric for damping performance is if $\tan\delta$ exceeds 0.3 over a temperature range of at least 60 °C[55]. Temperature sweeps are shown in Fig. 5d, and the pendant networks (Pend-3.5-25/75, Pend-5.8-25/75) meet this criteria. The telechelic network only exceeds 0.3 over a 43 °C window and is not as effective as the pendant counterparts in damping performance.

The mechanical properties of the mixed networks were evaluated through stress-strain curves from tensile tests conducted on Pend-3.5–100/0 and Pend-3.5-50/50 sample, one with only imine bonds and the other with 50/50 mixed bonds. In the imine network, the material demonstrates relatively poor mechanical property with low strain-at-break (16.5%). Although the mixed network demonstrates nearly the same stress at break, larger strain-at-break (95%) and higher toughness are displayed (Fig. 5d). This result agrees with Guan et al. in which the hydrogen bond is introduced into the dynamic covalent network. In that case, the hydrogen bonds play a role as sacrificial bonds to improve mechanical properties, which is also correlated with the energy dissipation[35]. Such a mechanism can also explain why the introduction of UPy into imine networks improves the overall mechanical properties.

### Two relaxation modes from UPy bond exchange

In mixed pendant PDMS networks, a high frequency shoulder adjacent to the fast main relaxation peak is observed (Fig. 6a, Supplementary 5a–c). The $G^*(\omega)$ spectra can be transferred to the continuous relaxation spectra $H(\tau)$ in the time domain[56] according to Eq. 2

$$G^{*}(\omega) = G'(\omega) + iG''(\omega) = \int_{-\infty}^{\infty} \frac{\omega\tau}{\omega\tau - i} H(\tau)\, d\ln\tau \qquad (2)$$

The higher frequency peak corresponds to a shorter timescale peak in the relaxation spectrum (Fig. 6b, Supplementary Fig. 5d–f). Such a shoulder, as well as the peak in $H(\tau)$ at shorter timescales, also appears in pendant PDMS with only UPy bonds (Fig. 6c, Supplementary Fig. 6a–c), i.e., on the high frequency side adjacent to the terminal relaxation process, indicating that the observed high frequency shoulder in the mixed sample originates purely from the UPy bond exchange, and is not impacted by the presence of imine crosslinkers. The timescale of the process beneath the high frequency shoulder

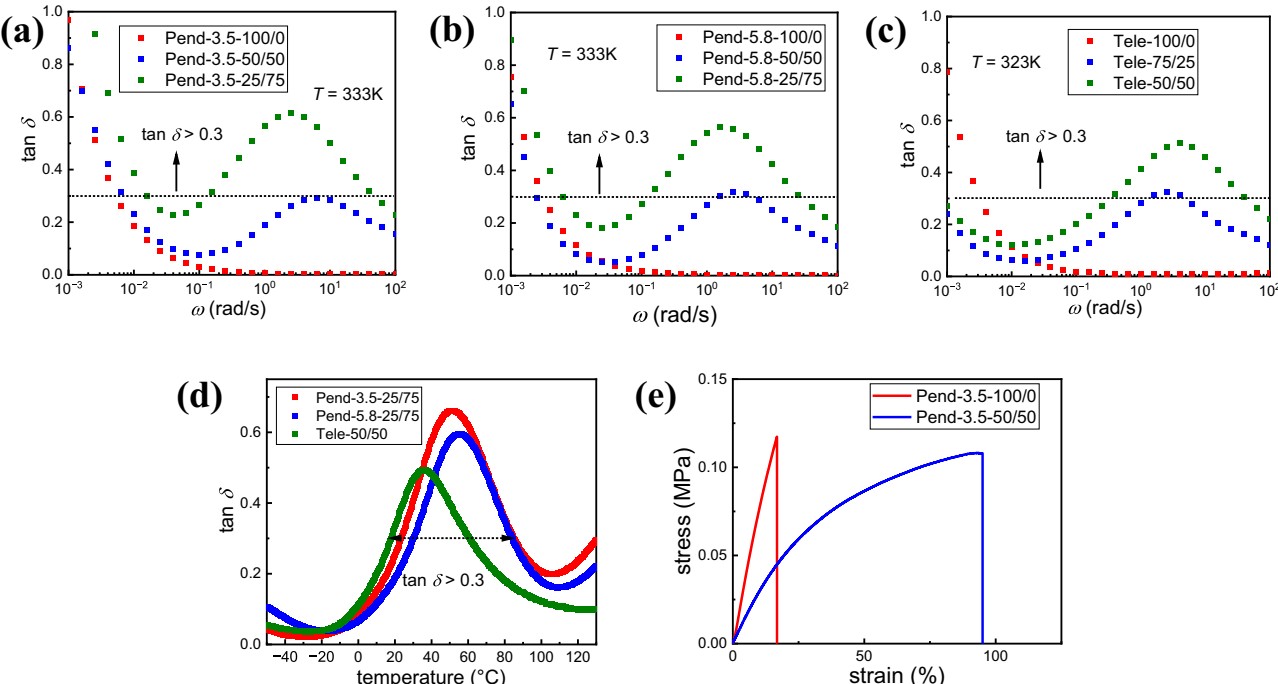

**Fig. 5 | Damping and mechanical properties of PDMS networks. a** $\tan\delta(\omega)$ spectra for Pend-3.5 samples with different bond ratios (different colors). **b** $\tan\delta(\omega)$ spectra for Pend-5.8 samples with different bond ratios (different colors). **c** $\tan\delta(\omega)$ spectra for telechelic samples with different bond ratios (different colors). **d** Temperature dependent $\tan\delta$ curves for pendant and telechelic networks. **e** Stress-strain curves for Pend-3.5-100/0 (red) and Pend-3.5-50/50 samples (blue).

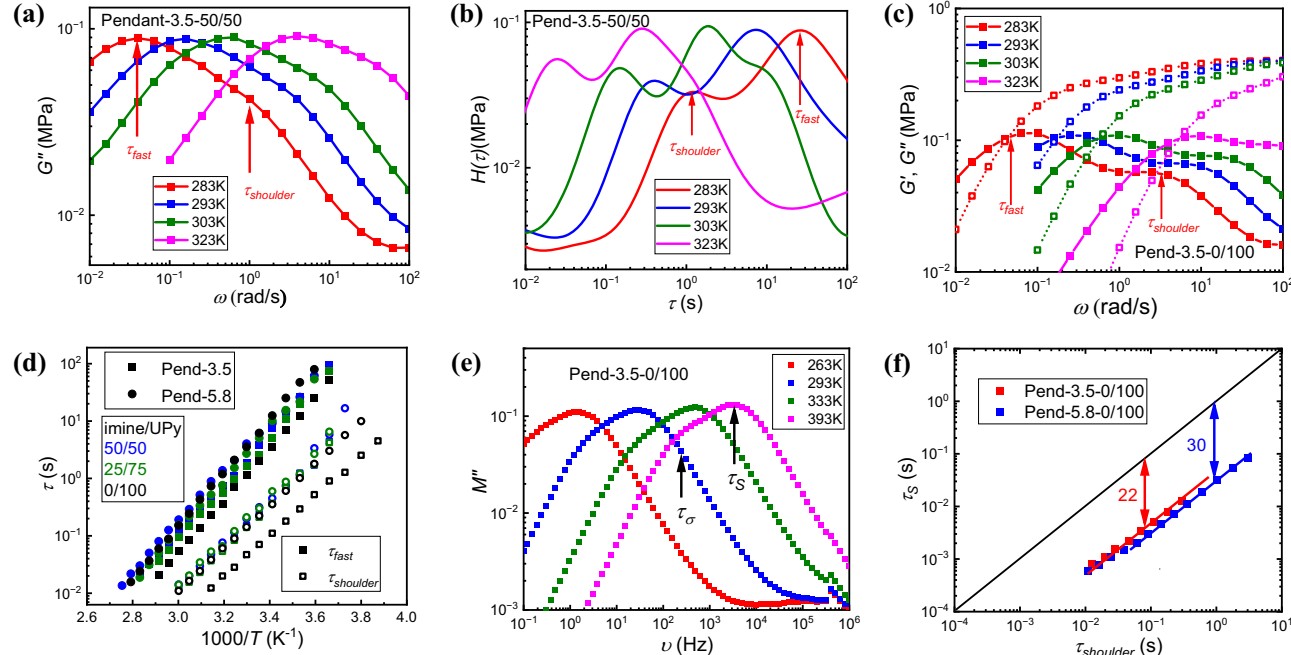

**Fig. 6 | Two relaxation modes from UPy bond exchange in pendant PDMS network. a** $G''(\omega)$ spectra of the fast process observed in Pend-3.5-50/50 at different temperatures (different colors) in which high frequency shoulder shows up. The position of $\tau_{fast}$ and $\tau_{shoulder}$ on the spectra measured at 283 K are labeled by arrows. **b** $H(\tau)$ spectra of the fast process observed in Pend-3.5-50/50 at different temperatures (different colors) in which high frequency shoulder shows up. The position of $\tau_{fast}$ and $\tau_{shoulder}$ on the spectra measured at 283 K are labeled by arrows. **c** $G'(\omega)$ (open symbol with dotted line), $G''(\omega)$ (closed symbol with solid line) spectra of the terminal relaxation process in Pend-3.5-0/100 sample at different temperatures (different colors) in which high frequency shoulder shows up. The position of $\tau_{fast}$ and $\tau_{shoulder}$ are labeled by arrows. **d** Temperature dependance of 2 adjacent relaxation processes ($\tau_{fast}$: closed symbol; $\tau_{shoulder}$: open symbol) correlated with UPy bond exchange in both Pend-3.5 (square) and Pend-5.8 (circle) PDMS with different bond ratios (different colors). **e** Dielectric $M''(\upsilon)$ spectra of Pend-3.5-0/100 at different temperatures (different colors). The position of $\tau_\sigma$ and $\tau_s$ are labeled by arrows. **f** Bond dissociation time with the timescale from high frequency shoulder (Pend-3.5-0/100: red; Pend-5.8-0/100: blue). The solid line represents an assumption that $\tau_s$ and $\tau_{shoulder}$ is equal.

($\tau_{shoulder}$) was acquired from the peak in $H(\tau)$ at shorter timescales. The mixed networks share the same $\tau_{shoulder}$ that is slightly slower than their pure UPy counterpart (Fig. 6d). The separation between $\tau_{shoulder}$ and the main fast relaxation process ($\tau_{fast}$) is between 1 and 2 orders of magnitude (Fig. 6d). The ratio of these two relaxation processes is shown in Table 3. The pendant PDMS with pure UPy bonds, which has the most UPy stickers, exhibits the largest separation between these two processes and was analyzed using the Sticky Rouse Model (SRM). The dynamics of polymers with sticky groups along the backbone can be described by the SRM which considers the repetitive breaking and reforming of the reversible bonds along the polymer backbone[15]. The SRM also predicts two relaxation modes where the faster one (with timescale of $\tau_1$) indicates a single bond exchange and major network rearrangement happens later (with timescale of $\tau_2$) after all the stickers along the backbone undergo bond exchange. These two predicted processes have a separation on a timescale determined by the number of sticky groups along the backbone $N_s$, and are related via $\tau_2 = \tau_1 N_s^2$. With a larger number of stickers along the chain, the separation of these timescales is anticipated to become bigger. This agrees with the separation of $\tau_{fast}$ and $\tau_{shoulder}$ of our pendant sample. Based on this experimental fact, we conclude that the high frequency shoulder (with the timescale of $\tau_{shoulder}$) corresponds to single UPy bond exchange,

whereas the main relaxation process at lower frequency ($\tau_{fast}$), indicates the rearrangement of UPy network.

To verify this hypothesis, dielectric spectroscopy was utilized to study the pendant PDMS network with only UPy bonds. In previous studies, bond dissociation of UPy was probed in the dielectric spectra of pendant networks, as the bond dissociation changes the overall dipole moment[57,58]. For pendant PDMS with only UPy stickers, the dielectric relaxation process correlated with UPy bond dissociation also shows up in the dielectric loss modulus (M'') spectra (Fig. 6e, Supplementary Fig. 7a, 7b). The low frequency process indicates conductivity relaxation[59], whereas the high frequency process indicates the bond dissociation of UPy. To acquire the relaxation time of these two processes, i.e., $\tau_s$ for UPy bond dissociation and $\tau_\sigma$ for conductivity relaxation. The M'' spectra were analyzed through Havriliak–Negami equation[60] and the acquired timescales were labeled with arrows (Fig. 6e, Supplementary Fig. 7a, b). The detailed analysis method is described in the Methods section.

The acquired $\tau_s$ at various temperatures is faster than the high frequency shoulder process from rheology for both pendant PDMS with pure UPy bonds, with a separation ~ 22 for Pend-3.5-0/100 sample and a separation ~ 30 for Pend-5.8-0/100 sample (Fig. 6f, Supplementary Fig. 8). A similar magnitude of separation was also observed for polymers with UPy stickers[57], attributed to the bond exchange mechanism for UPy stickers. After sticker dissociation, an open sticker will reassociate with its original partner several times before it diffuses to a new partner to form a new dynamic bond[61,62]. In other words, bond exchange can be several times longer than the bond dissociation time for a single UPy sticker, which results in the contrast of rheological and dielectric timescales. The dielectric results support the hypothesis that $\tau_{shoulder}$ corresponds to the dissociation of a single UPy motif.

**Table 3 | Ratio of $\tau_{fast}/\tau_{shoulder}$ for both pendant and telechelic PDMS with different imine/UPy ratios**

|          | 75/25 | 50/50 | 25/75 | 0/100 |
|----------|-------|-------|-------|-------|
| Pend-3.5 | N/A   | 12    | 14    | 31    |
| Pend-5.8 | N/A   | 14    | 18    | 25    |
| Tele     | 21    | 12    | 12    | NA    |

A high frequency shoulder adjacent to the main fast relaxation mode also appears in the $G''(\omega)$ spectra of telechelic mixed networks (Fig. 7a, Supplementary Fig. 9a, b), even for Tele-25/75 which only shows a fast mode (Fig. 6a). To acquire the timescale of the high frequency shoulder ($\tau_{shoulder}$), the $G''(\omega)$ spectra were also transferred to H(τ) according to Eq. 2 (Fig. 7b, Supplementary Fig. S9c, d) and $\tau_{shoulder}$ was acquired from the peak of the shorter timescale. In pure UPy networks, however, only one relaxation process shows up which is correlated to the UPy bond exchange that leads to terminal relaxation (Supplementary Fig. 9e, f). With only UPy motifs, the telechelic chains undergo head-to-tail association through the binary interaction of different UPy stickers so that telechelic chains only form effectively linear architectures[8,61]. In this case, bond exchange results in significant network rearrangement and the terminal mode shows up[63]. The experimental results from shear modulus spectra indicate the 3-arm imine crosslinker is essential for two adjacent relaxation modes in telechelic PDMS, even for the case that imine network cannot percolate throughout materials (i.e., Tele-25/75 sample) where the structure is a 3-arm star-shaped hyper branch end-functionalized by UPy (Fig. 7d). The center of the star is the imine crosslinker. To understand why UPy exchange results in two adjacent relaxation modes in mixed telechelic network, we consider that the existing imine crosslinker acts as if it was a permanent junction since imine bond exchange is orthogonal to and much slower than the UPy bond exchange. Existing theory for star-like polymers[64,65] predicts two relaxation modes due to the architecture. The high frequency process indicates arm relaxation, whereas the low frequency one indicates the diffusion of the star-shaped polymer which results in major network rearrangement. Such hierarchical dynamics due to the 3-arm architecture has also been investigated in star-shaped polymers with reversible bonds as well. One example of these two adjacent relaxation modes was observed in 3-arm telechelic polyisoprene end-functionalized by zwitterions[66]. The separation of the timescales of single bond exchange and network rearrangement was found in star-shaped polymers with metal-ligand associations as well[67], around one order of magnitude.

Figure 7c shows the temperature dependence of $\tau_{shoulder}$ and the $\tau_{fast}$. All the mixed networks share the same $\tau_{shoulder}$ which is slightly slower than $\tau_{fast}$ of the pure telechelic UPy network, demonstrating that $\tau_{shoulder}$ represents the bond exchange of UPy groups in the mixed network. The separation between the fast main process $\tau_{fast}$ and $\tau_{shoulder}$ is also slightly larger than one order of magnitude (Fig. 7c and Table 3), which agrees with the theory prediction and experimental results for star-like polymers[66–68]. Thus, the adjacent relaxation process in telechelic mixed network originates from the 3-arm architecture formed by the imine crosslinker. The high frequency shoulder indicates single bond exchange and the main fast relaxation process results from major network rearrangement due to the diffusion of the star-shaped structure. Even in the percolated imine network, there still exists a small fraction of 3-arm hyperbranched structures (Fig. 7d). The UPy bond exchange can still result in partial network rearrangement once the 3-arm architecture is able to diffuse in the mixed network so that the two adjacent relaxation processes due to UPy exchange still

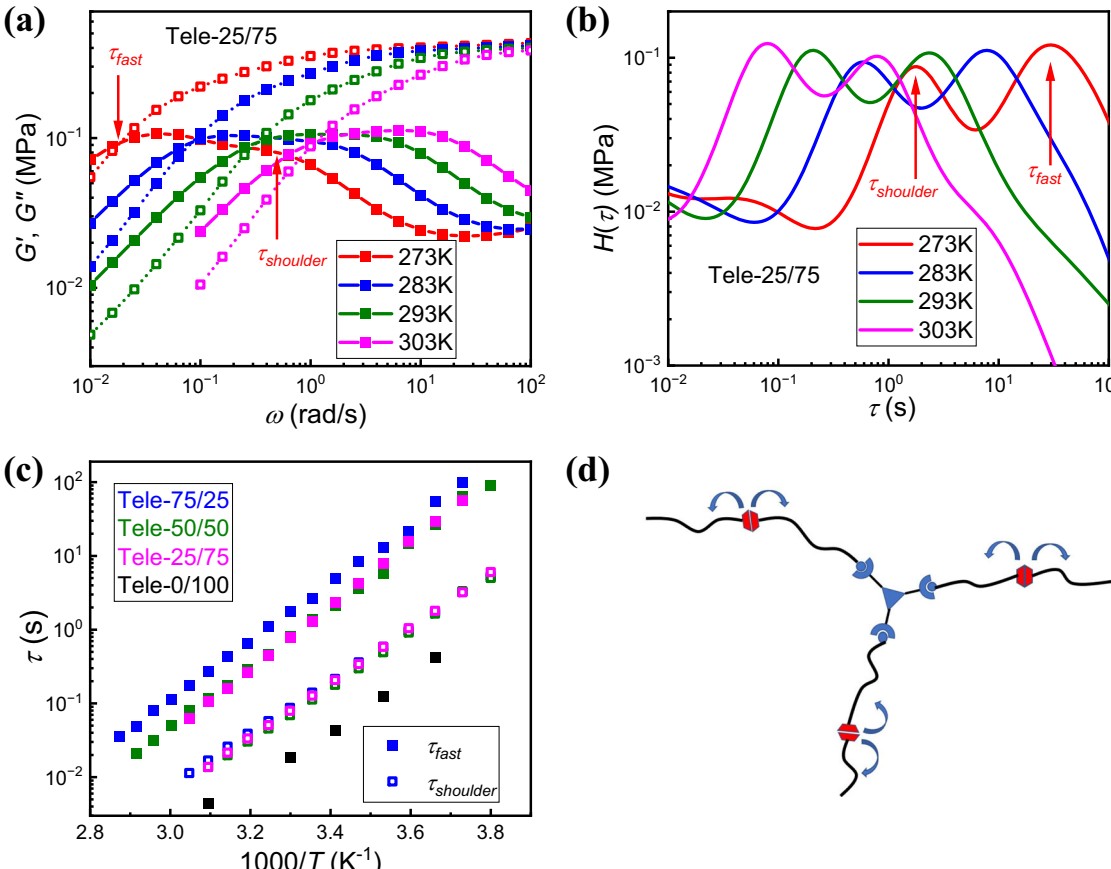

**Fig. 7 | Two relaxation modes from UPy bond exchange in telechelic PDMS network. a** $G'(\omega)$ (open symbol with dotted line), $G''(\omega)$ spectra (close symbol with solid line) of the terminal relaxation process in Tele-25/75 sample at different temperatures (different colors) in which high frequency shoulder shows up. The position of $\tau_{fast}$ and $\tau_{shoulder}$ on the spectra measured at 273 K are labeled by arrows. **b** H(τ) spectra of the terminal relaxation process in Tele-25/75 at different temperatures (different colors) in which high frequency shoulder shows up. The position of $\tau_{fast}$ and $\tau_{shoulder}$ on the spectra measured at 273 K are labeled by arrows. **c** Temperature dependance of 2 adjacent relaxation processes ($\tau_{fast}$: closed symbol; $\tau_{shoulder}$: open symbol) correlated with UPy bond exchange in telechelic PDMS with different bond ratios (different colors). **d** Cartoon of the star-like structure formed by imine bond below the percolation threshold.

prevail (Supplementary Fig. 9a–d). The unraveled physics behind the main fast relaxation process also explains why the introduction of the imine crosslinker retards the UPy bond exchange for telechelic mixed network (Fig. 4b, c).

## Discussion

In this work, the presence of multiple relaxation modes in pendant and telechelic PDMS is achieved by introducing two orthogonal dynamic crosslinkers forming quadrupole hydrogen bonds and dynamic covalent imines. Two well-separated relaxation modes in the mixed network are exhibited regardless of pendant or telechelic architecture. The predominant factor is whether the transient network formed by imine bonds can percolate throughout materials. If well-separated relaxation modes show up, the low frequency plateau modulus can be tuned by the imine/UPy ratio. In pendant PDMS with mixed bonds, the timescale of both fast and slow relaxation process generally agrees with the bond exchange timescale observed from pure UPy and imine network, respectively. In the case of telechelic PDMS, however, mixed networks demonstrate a significantly longer timescale of the fast and slow relaxation modes compared with their pure counterparts. For the mixed network, exceptional wave damping property and improved mechanical properties are also demonstrated arising purely from bond exchange and not $T_g$.

The bond exchange of UPy groups in mixed PDMS with both pendant and telechelic networks results in two adjacent relaxation processes, demonstrated as a high frequency shoulder besides the main fast relaxation process. A separation around 1-2 orders of magnitude is observed between their timescales. With both backbone architectures, the high frequency shoulder corresponds to single UPy bond exchange. In pendant PDMS mixed network, the main fast relaxation process is slower as the UPy network gets rearranged after all UPy stickers along the polymer backbone undergo bond exchange. In telechelic PDMS mixed network, the presence of 3-arm imine crosslinkers makes the network rearrangement due to UPy bond exchange slower, retarding the main fast relaxation process.

Overall, up to three relaxation modes are demonstrated in our designed PDMS network with both hydrogen bonding and dynamic covalent transient bonds which have orthogonal exchange mechanisms. This research provides a promising method of rational design of damping material with well separated relaxation modes through controlling the network architecture and ratio of orthogonal dynamic bonds, which will also have broad interests for design of other advanced materials with dynamic bonds.

## Methods
### Materials Synthesis
**Synthesis of telechelic and pendant PDMS-UPy**. To synthesize telechelic PDMS-UPy, 2-(1-Imidazolylcarbonylamino)-6-methyl-4-[1H]-pyrimidinone (UPy-CDI) was synthesized firstly, the detailed synthesis method is mentioned in Section 11 of supplementary information with synthesis route (Supplementary Fig. 10a) and IR spectra result (Supplementary Fig. 10b). After UPy-CDI was successfully synthesized, excess UPy-CDI was suspended in chloroform and then added into telechelic PDMS-NH$_2$ (1 g) solution. The reaction was conducted at 353 K for 16 hours under nitrogen (Supplementary Fig. 11a). After that, the mixture was cooled to room temperature. The excess UPY-CDI was still suspended in the solution so that vacuum filtration with a glass frit and filter paper was applied to remove the unreacted UPy-CDI. The solution was then washed three times with brine to remove the imidazole before being dried under vacuum at room temperature overnight. The final product was characterized by NMR. A typical $^1$H-NMR result is shown in Supplementary Fig. 11b, indicating the successful linkage of the UPy block on the PDMS backbone. To synthesize pendant PDMS-UPy, the procedure is identical to that for telechelic PDMS.

**Synthesis of telechelic and pendant PDMS with imine group**. To synthesize pendant PDMS imine networks, terephthalaldehyde was used as the imine crosslinker. Terephthalaldehyde and PDMS-NH$_2$ (1 g) were dissolved in chloroform and reacted for 12 hours under nitrogen (Supplementary Fig. 12a). Then chloroform was allowed to slowly evaporate. The sample was further dried in vacuum at 393 K for 24 hours to remove the residual chloroform and the water formed during synthesis. The imine bond formation was verified by the IR spectra. The peak at 1644 cm$^{-1}$ indicates the vibration of the imine bond (Supplementary Fig. 12b). To synthesize telechelic PDMS imine networks, benzene-1,3,5-tricarbaldehyde was used as the crosslinker. The other steps are the same as the those for pendant PDMS with imine groups.

**Synthesis of PDMS with mixed network**. To synthesize the mixed network, the UPy group was first introduced to the PDMS network. A fraction of the NH$_2$ groups in telechelic PDMS or pendant PDMS was reacted with the UPy-CDI, following the same steps used to synthesize PDMS with only UPy groups. Supplementary Fig. 13 indicates the $^1$H-NMR result of a typical pendant PDMS chain partially functionalized by UPy (50% of NH$_2$ get functionalized by UPy). After that, the partially functionalized PDMS was dissolved in chloroform and terephthalaldehyde or benzene-1,3,5-tricarbaldehyde was added. The following step was the same as the synthesis of PDMS imine network.

### Gel permeation chromatography (GPC) measurement
The relative molecular weights of pendant and telechelic Boc-protected PDMS precursors were determined using a Tosoh EcoSEC Elite Model HLC-8420, equipped with two Tosoh TSKgel Alpha-M columns and a Tosoh Dual-Flow RI detector. The elution solvent was HPLC grade chloroform. The PDMS precursors were dissolved at room temperature and filtered through 0.45 μm Teflon syringe filters (Fisher Scientific). The sample concentrations were ~4 mg/mL, and the injection volumes were 50 μL. The flow rate was 0.6 mL/min, and the column temperature was 40 °C. Several narrow polydispersity index (PDI) Polystyrene (PS) standards were used for conventional calibration. The data were processed using Tosoh SECview software.

### Differential scanning calorimetry (DSC) measurements
DSC measurements were employed to probe the glass transition temperature ($T_g$) of the samples using a Q2500 DSC equipment from TA Instruments. The samples were dried in a vacuum oven overnight before being placed into DSC pans. The samples were first equilibrated isothermally at 120 °C for 5 minutes to remove the thermal history before being quenched to −180 °C (to avoid crystallization). After equilibration for 5 minutes, the samples were heated to 120 °C at a rate of 10 °C /min. This procedure was repeated twice for each sample to ensure repeatability.

### Shear Rheology
Small-amplitude oscillatory shear (SAOS) was utilized to probe the viscoelastic properties of the samples using a DHR-2 (TA Instruments). To prepare the samples for rheology, the samples were placed into an 8 mm diameter disk mold and hot pressed at 110 °C for 5 – 10 minutes to ensure a uniform sample. Then the sample was taken out of the mold and placed on the rheometer using parallel plate geometry with a disk diameter of 8 mm. For the frequency sweep test, the experiments were conducted with an angular frequency range from $10^2$ to $10^{-4}$ rad/s at a variety of temperatures range from −30 to 120 °C. For the temperature sweep, the experiments were conducted from −50 to 130 °C with a frequency of 1 rad/s. Since most of the measurement was conducted in the rubbery plateau regime, the strain amplitude during the measurement was chosen to be 3%.

**Broadband Dielectric Spectroscopy (BDS) Measurement and data analysis for dielectric modulus spectra**

Then BDS spectra in the frequency range from $10^{-1}$ to $10^6$ Hz were measured using a Novocontrol system that includes an Alpha-A impedance analyzer and a Quatro Cryosystem temperature control unit. The dielectric measurement was conducted in a parallel-plate dielectric cell made of stainless steel with an electrode diameter of 10 mm, and capacitance 3.3 pF with an electrode separation of 210 μm. The liquid-like pendant PDMS with pure UPy network was loaded into the dielectric cell at 120 °C. The measurement was conducted from 120 to −30 °C, in 5 °C intervals. After each temperature change, the samples were equilibrated for 10 min to reach thermal stabilization within 0.1 °C.

The dielectric $M''$ spectra contains two relaxation processes in the same frequency window and the conductivity relaxation process has a Debye-like shape[59]. Thus, the dielectric spectra were analyzed using the equation with an addition of an Havriliak–Negami function[60] plus a Debye function representing two relaxation processes (Eq. 3)

$$M''(v) = Im\left\{ \frac{\Delta M_s}{[1+(2\pi i v \tau_{s,HN})^\alpha]^\beta} + \frac{\Delta M_\sigma}{1+2\pi i v \tau_\sigma} \right\} \tag{3}$$

in which $\Delta M_s$ and $\Delta M_\sigma$ are the "modulus strength" of UPy bond dissociation and conductivity relaxation, respectively. $\tau_{s,HN}$ and $\tau_\sigma$ are the HN relaxation time for UPy bond exchange and characteristic relaxation time of conductivity relaxation process. $\alpha$ and $\beta$ are the stretching parameters for the UPy bond dissociation process correlated with peak broadening and asymmetry, respectively. All the parameters mentioned above are fit parameters. The contribution of each process to the dielectric spectra is reflected by dashed lines in Fig. S8. The characteristic timescale for the UPy bond exchange ($\tau_s$) was transferred from $\tau_{s,HN}$ according to Eq. 4[69].

$$\tau_s = \tau_{s,HN} \left[ \sin\left(\frac{\alpha\pi}{2+2\beta}\right) \right]^{-\frac{1}{\alpha}} \left[ \sin\left(\frac{\alpha\beta\pi}{2+2\beta}\right) \right]^{\frac{1}{\alpha}} \tag{4}$$

**Small angle X-ray scattering (SAXS) measurement**

SAXS measurement was conducted on a small angle X-Ray scattering system with Pilatus 300 Detector and Cu Kα as radiation source with wavelength of 1.54 Å. The distance between the sample and detector was 1.36 m. The exposure time is 300 s and the intensity of the direct beam is 28175 counts per second. The sample was placed perpendicular to the X-ray beam. The X-ray measurements were performed at room temperature.

**Tensile test**

The tensile test was conducted in TA Instruments Q800 Dynamic Mechanical Analyser (DMA) with strain-controlled mode at room temperature. The sample was made into a rectangular stripe with the dimensions of 20 mm × 5 mm × 0.6 mm. The strain rate was set to 1% per minute.

## Data availability

The authors declare that full experimental details and characterization of materials are available in the Supplementary Information. All raw data that support the findings in this study are available from the corresponding author upon request.

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

## Acknowledgements

The authors gratefully acknowledge support from the Air Force Office of Scientific Research (AFOSR) under support provided by the Organic Materials Chemistry Program (grant FA9550-20-1-0262 to S.G. and C.M.E.). Portions of the work were also supported by the National Science Foundation through award CBET-2029928 (X-ray scattering of vitrimers, to Y.T. and C.M.E.). Aspects of this work were performed at the Materials Research Laboratory and School of Chemical Sciences facilities in UIUC.

## Author contributions

S.G. designed and synthesized the polymer network, and performed the DSC, rheology, dielectric measurements. Y.T. performed the X-ray scattering measurements. C.M.E. conceived of and supervised the project. All authors contributed to writing of the manuscript.

## Competing interests

The authors declare no competing interests.
