## [Peer Review File · Nature Communications]

Polymer Architecture Dictates Multiple Relaxation Processes in Soft Networks with Two Orthogonal Dynamic BondsReviewers' Comments:

Reviewer #1:

Remarks to the Author:

This manuscript describes mechanical relaxation of polysiloxane (PSX) networks containing either or both reversibly associating hydrogen bond groups and/or vitrimeric dynamic crosslinks. The emphasis is on architecture-property relationships. Various multimodal relaxations were observed that depend on the placement of dynamic functional groups (pendent versus telechelic) or the mole fraction of dynamic functional group type (H-bonding versus dynamic covalent). The results of the study mesh fairly well with existing literature and offer new insight into the engineering of dynamic networks with tunable viscoelastic properties. The overall organization and structure of the paper is appropriate. The discussion is well written and the data support the authors explanation of what is going on quite well. I did not see any obvious flaws in the authors' reasoning. While I'm generally supportive of publication, there are a few items that the authors are encouraged to clarify in a revision:

(1a) When describing the composition of functionalized PDMA, the definition of 'amine percentage' needs to be clear-- mass percentage, mole percentage, monomer percentage? Errors in these measurements are important to understand because they may propagate into data interpretations, so the composition of the materials should be determined as accurately as possible.

(1b) when synthesizing the telechelic and pendent PDMS-UPy: how was the material filtered? Did a precipitate form, or was the suspension filtered directly? Maybe describe the type of filter to help people reproduce the experiments.

(2) For DSC results, the T_g may vary depend on composition, but this variation may be beneath the detectable range, so I would refrain from flat-out stating that there is no dependence on the Imine/UPy composition. Also, it looks like endotherms are downward, and this should be stated in the Figure caption or shown in the figure.

(3) The authors should also be careful by somewhat cavalierly concluding that the lack of SAXS scattering indicates the lack of microphase segregation. I'm presuming that a relatively low flux, benchtop unit was used, and it is not clear how thick the samples were or how long they were irradiated. In some weakly phase-segregated systems, long exposure times or high flux tools are needed to see scattering.

(4) Our group previously used hydrosilylation to functionalize UPy onto PDMS chains (Adv. Funct. Mater. 2019, 1903721), and we studied rheology of the formed networks as well. Stress relaxation studies showed evidence that stress lowers the activation energy for UPy dissociation. I imagine similar observations could be made on this system. Our manuscript examined pendent UPy's on a covalent network that, if the authors desire, could be comparable to the mixed-mechanism system in the current study. However, I believe only a single relaxation mode was observed in our study.

(5) Several times the connectivity (percolation) of the network is discussed. It would be good to know exactly where this occurs. Flory-Stockmeyer theory shows the gel point conversion to be $X = (1/(r(f_a - 1)(f_b - 1)))$ where f_a and f_b are the functionality of monomers (two and three for the current system) and r is the stoichiometric ratio of the components. I believe this equation can be applied to estimate the theoretical gel point at a critical concentration of trimers.

(6) The phrase 'orthogonal' is used to describe the dynamic bonds, and, presumably it infers that the dynamics are independent of one another. Was this verified?

There are a handful of minor issues listed below.

Minor issues:

- 'materials' should be removed as the last word of the first sentence in the abstract because it is not an 'application'.
- 'hydrogen-bond' should be hyphenated throughout to improve readability.
- LN 16 was confusing the first time I read it. After reading the paper it made sense, but maybe rewrite it.
- I believe Table 1 caption should read 'amine' instead of 'imine'
- LN 28: maybe 'viscoelastic spectra is the PRESENCE OF loss peaks...'
- LN 62: The last sentence in this paragraph containing 'many key questions remain...' leaves the reader hanging and wondering what these questions are. Maybe just rephrase it.
- LN 105: "pendent and telechelic PDMS with a single dynamic bond TYPE were synthesized'

Mitch Anthamatten

Reviewer #2:

Remarks to the Author:

The contribution from the Evans group focuses on relaxation and dynamics in multiply dynamic materials and crosslinkers. The effect of architecture is explored through both pendant and telechelic polymers. The work is an important contribution to the knowledge and design of multiply dynamic materials. However, in its current form it is not suitable for Nat Comm. However, it could be improved to meet the requirements for this generalist journal.

Specific issues to address

1. Currently the work is too physics oriented. It is not easily accessible for those more on the chemistry end of dynamic materials. Diagrams like Figure 1 and Figure 6D are helpful, but it would be stronger if they also existed in between to help the chemistry focused readers better understand the work
2. The definition of the t_{fast} is not clear. The reviewer can certainly "eyeball" where it should be, but a greater and expanded definition and explanation of it needs to be given. The t_{slow} would be an improvement. It is clearly related to Eq 2, and the generated H function, but again the description is too physics heavy for the broad readership of Nat Comm
3. Some of the activation energies in Table 2 are extraordinarily small. 10 kJ/mol for imine exchange is very low, and indicates virtually no temperature dependence. Indeed these values seem very low compared to literature values of imine exchange E_a . See Coates et al. 70 kJ/mol (DOI: 10.1039/C9PY01957J). It was only in the presence of very highly withdrawing groups (sulfone) that are quite different from the system studied here which should be much less with drawing. Further analysis and comparison of these results with respect to the literature would be highly beneficial.
4. The overall results are interesting but there appears to be several segments of the literature that are missing. For instance prior work on dynamic materials and architectural effects (E.g. work of Sumerlin, Konkolewicz, and Garcia including reviews) have been completely neglected. Similarly, much of the work in multiply dynamic materials has not been cited (Lehn, Guan etc) including several reviews from the past ~4 years .
5. How effective is water removal? Clearly for imines, water needs to be removed, and varying amounts of water can introduce different exchange mechanisms.
- 6 Minor issue but sometimes the terminology changes from Pendent to Pendant (Including some figures)

Reviewer #3:

Remarks to the Author:

In the manuscript "Polymer architecture dictates multiple relaxation processes in soft networks with two orthogonal bonds" Evans and co-workers build upon their expertise in the mechanics of covalent adaptable networks by introducing orthogonal dynamic moieties into commercially-available PDMS variants. As opposed to earlier work from the group (ref 19), here the mixture of fast and slow bonds result in resolvable relaxation modes with strong frequency separation (~ 4 orders of magnitude). Overall, the work is sound and comprehensive from the perspective of synthesis and characterization, but I believe it lacks the novelty and the impact for the broad readership of Nature Communications (however, this article would be well-suited for publication in a specialized polymers journal). As such, I do not recommend it for publication in Nature Communications.

I have broken my comments into Novelty, Major, and Minor below

Novelty:

1) In the introduction of this work, the authors describe the utility of complex viscoelastic behaviors in a suite of applications (tissue engineering, energy dissipation, adhesion, additive manufacturing, and overall mechanical properties), but the authors do not demonstrate the utility of the reported materials in any of these applications. To attain adequate novelty and impact, a demonstration of the reported materials in use for any of the described applications should be carried out.

2) Similarly, as materials are often applied (or at least spend a significant amount of time between activity) under static or pseudo-static conditions at or around room temperature, it may be beneficial to include some general tensile characterization to more easily benchmark these materials.

3) At several points in the manuscript, the authors note that the key aspects or synthetic design strategies reported in this work have been observed/used in previous reports

a) Two relaxation peaks from supramolecular metal-ligand-based materials (Ref 16)

b) Appearance and disappearance of multiple relaxation modes in dynamic materials based on PDMS (Ref 20)

c) Pendant UPy stickers in combination with dynamic covalent moieties (Ref 25)

d) High frequency should relaxation mode in UPy stickers measured via dielectric spectroscopy (Refs 44,45)

e) The most significant overlap comes from reports that have prepared and investigated telechelic PDMS based materials with a mixture of imine and UPy motifs (Ref 28). Granted, this report did not carry out detailed analysis of the rheological properties.

In its current state, this work serves as an incremental advance in the design of complex viscoelastic materials as the synthesis, design strategy, and overall phenomena have been previously reported.

That being said, the authors have put together a very thorough investigation with well-thought out experiments and analysis, it just lacks the novelty required for a broad readership.

The authors wisely state that "... mixing fast and slow bonds does not guarantee well-separated relaxation modes in polymer networks" which sets up an interesting problem in determining design rules for developing such materials. However, this work does not supply sufficient substrate scope or breadth for the rational design of such behavior beyond the prepared materials.

Major:

4) Overall, this manuscript can be difficult to parse. I believe this is, in part, due to a lack of naming scheme for the various materials being studied. I strongly recommend that the authors find a short and legible means of getting across the architecture, amine percentage, and dynamic moiety ratio for easier reading.

5) As a control, a few materials containing only 50% imine bonds (leaving out the 50% UPy) should be prepared to highlight the lack of faster relaxation modes.

6) Why is the activation energy of the fast mode (as shown in Figure 4b) not analyzed in Table 2 (for the 75/25 and 50/50 samples)? It appears to be similar in magnitude to the highest E_a measured for

the imine bond exchange. Is this sensible?

Minor:

7) Figure 1 could use a reaction scheme to show the final chemical structure of the installed UPy groups after reaction with the amine. Also, Figure 1 is quite large and pixelated, and could easily get across the same information in half the space.

8) The legends of certain figures are quite close to data points, which can cause confusion. I suggest surrounding the legends with a box to help separate the legends from data points. A particularly egregious example can be seen in Figure 3c, where the pendant sample legend is effectively collinear with the reported data.

9) The authors went through the trouble of taking an NMR to verify the amine loading, but did not do an independent analysis of the molecular weight of these commercial polymers (which in some cases can deviate significantly from the supplier report).

Response to the reviewers

Reviewer 1:

(1a) When describing the composition of functionalized PDMA, the definition of 'amine percentage' needs to be clear-- mass percentage, mole percentage, monomer percentage? Errors in these measurements are important to understand because they may propagate into data interpretations, so the composition should of the materials should be determined as accurately as possible.

We thank the reviewer for pointing this out and we have clarified our wording. The amine percentage in this manuscript is molar percentage with respect to the repeat units of PDMS. We have modified “amine percentage” to “molar percentage of amine” throughout the manuscript.

(1b) when synthesizing the telechelic and pendent PDMS-UPy: how was the material filtered? Did a precipitate form, or was the suspension filtered directly? Maybe describe the type of filter to help people reproduce the experiments.

After the synthesis, the excessive UPY-CDI was still suspended in the solution as it cannot dissolve in chloroform. Then, vacuum filtration was used with a glass frit and filter paper to remove the excessive UPY-CDI, which is confirmed by NMR. Following the reviewer’s comment, we modified the synthesis procedure of PDMS-UPy on Page 26 of the manuscript.

(2) For DSC results, the T_g may vary depend on composition, but this variation may be beneath the detectable range, so I would refrain from flat-out stating that there is no dependence on the Imine/UPy composition. Also, it looks like endotherms are downward, and this should be stated in the Figure caption or shown in the figure.

The table below indicates the T_g of all the dynamic networks involved in this research. Thus, from DSC measurement, we do not see very significant change of T_g with different compositions. The T_g is mainly correlated with the initial NH₂ molar percentage which determines the crosslink density.

Sample	T _g (C)
Tele-0/100	-123.9
Tele-25/75	-123.8
Tele-50/50	-122.8
Tele-75/25	-123.0
Tele-100/0	-122.9
Pend-3.5-0/100	-121.9
Pend-3.5-25/75	-121.01
Pend-3.5-50/50	-120.01
Pend-3.5-100/0	-119.73
Pend-5.8-0/100	-116.8
Pend-5.8-25/75	-115.82
Pend-5.8-50/50	-115.15

We modified the sentence to “but it does not discernibly vary with the different compositions of dynamic bonds” on Page 7. In addition, we added the table above in SI (table S1).

(3) The authors should also be careful by somewhat cavalierly concluding that the lack of SAXS scattering indicates the lack of microphase segregation. I'm presuming that a relatively low flux, benchtop unit was used, and it is not clear how thick the samples were or how long they were irradiated. In some weakly phase-segregated systems, long exposure times or high flux tools are needed to see scattering.

We have added details of the X-ray setup to the experimental methods section, and the benchtop system at our university has high flux and a high sensitivity detector. To verify that the measurement conditions can detect microphase separation, we synthesized the telechelic PDMS-UPY with an additional urea group between the PDMS backbone and the UPy group. This kind of telechelic polymer was reported to show microphase separation (Journal of Polymer Science Part A: Polymer Chemistry, 46(12), 3877-3885.). The chemical structure is shown below:

The SAXS spectra was measured with the same conditions and thickness as the two samples reported in the manuscript. We can see a clear peak at 0.7 nm^{-1} , indicating microphase separation which is not present in the current samples. We have added this information to the manuscript and the Supporting Information.

This SAXS result indicates that with the same condition, microphase separation can be measured on our X-ray scattering machine. Thus, we can conclude that in sample Pend-3.5-0/100 and Pend-3.5-25/75, there is no visible microphase separation through the X-ray scattering measurement.

In the manuscript, we have added the exposure time and the intensity of the direct beam into the Methods section (Page 30 of the manuscript). We have modified the conclusion of SAXS to be “the absence of visible microphase separation by SAXS” on Page 9. Also, we modified the title of the X-ray scattering result to be “Absence of Visible Microphase Separation”.

(4) Our group previously used hydrosilylation to functionalize UPy onto PDMS chains (Adv. Funct. Mater. 2019, 1903721), and we studied rheology of the formed networks as well. Stress relaxation studies showed evidence that stress lowers the activation energy for UPy dissociation. I imagine similar observations could be made on this system. Our manuscript examined pendant UPy's on a covalent network that, if the authors desire, could be comparable to the mixed-mechanism system in the current study. However, I believe only a single relaxation mode was observed in our study.

We thank the suggestion from the reviewer of comparing the results in this manuscript with previously published results. In terms of whether stress lowers the activation energy for UPy dissociation, this suggestion was very interesting, and we agree that stress might affect activation energy. However, in this work we are focusing on linear regime and trying to understand the linear response of the dynamic polymer. It would be interesting for future work to look at lowered activation energies in the case of the mixed system, but is beyond the scope of the present manuscript.

In terms of the comparison of shear modulus, we extracted the shear modulus master curve of XDN-25k-1.6 from Figure 3b in the AFM paper and compared it with the spectra of one of our pendant PDMS samples (Pend-5.8-50/50) (see the graph below). The loss peak correlated with UPy bond exchange was labeled with an arrow. The result indicates that the loss peak of XDN-25k-1.6 generally agrees with our result, indicating the consistency of the UPy bond exchange timescale. Since the sample in the AFM paper has only one type of dynamic crosslinker, only a single relaxation mode is expected to show up. We used the following figure to demonstrate that the fast relaxation process in our mixed sample originates from UPy bond exchange (Page 10, Figure S4).

(5) Several times the connectivity (percolation) of the network is discussed. It would be good to know exactly where this occurs. Flory-Stockmeyer theory shows the gel point conversion to be $X = (1/(r(fa-1)(fb-1)))$ where fa and fb are the functionality of monomers (two and three for the current system) and r is the stoichiometric ratio of the components. I believe this equation can be applied to estimate the theoretical gel point at a critical concentration of trimers.

We applied Flory–Stockmayer Theory to study when the percolation (gelation) will happen. In our study, we found that the percentage of NH_2 reacting with trialdehyde to form imine bond on the telechelic PDMS needs to reach 50% percent so that the percolation (gelation) of the imine network can occur. A detailed description of how we get this percolation threshold has been added in SI.

We have also modified the main text of the manuscript, mentioning that the percolation threshold is estimated by Flory–Stockmayer Theory.

(6) The phrase 'orthogonal' is used to describe the dynamic bonds, and, presumably it infers that the dynamics are independent of one another. Was this verified?

By “orthogonal” we mean that the dynamic bond exchange mechanism is independent of each other. UPy groups do not cause imine exchange, and imines do not cause UPy dissociation. However, this does not mean the dynamics are necessarily independent if multiple dynamic bonds exchange along the same chain. In our work, the invariance of the exchange timescales for both bonds in the pendant networks supports the picture that the bond dynamics are not impacting each other. In the telechelic systems, the relaxation of UPy is slightly slowed but this is attributed to the network architecture because of the pendant network results. We have clarified our definition of “orthogonal” in the manuscript on Page 4.

Minor issues:

- *'materials' should be removed as the last word of the first sentence in the abstract because it is not an 'application'.*

We have modified the abstract accordingly (Page 1).

- *'hydrogen-bond' should be hyphenated throughout to improve readability.*

We have modified this term throughout.

- *LN 16 was confusing the first time I read it. After reading the paper it made sense, but maybe rewrite it.*

We have rewritten that sentence to improve the clarity:

“A hydrogen-bonding group and a vitrimeric dynamic crosslinker are combined into the same network, and multimodal relaxation is observed in both pendant and telechelic networks” (Page 1)

- *I believe Table 1 caption should read 'amine' instead of 'imine'*

We have modified Table 1 accordingly (Page 6).

- *LN 28: maybe 'viscoelastic spectra is the PRESENCE OF loss peaks...'*

We have modified the sentence accordingly (Page 2).

- *LN 62: The last sentence in this paragraph containing 'many key questions remain...' leaves the reader hanging and wondering what these questions are. Maybe just rephrase it.*

We have modified the sentence accordingly.

“The roles of bond exchange mechanisms and network architecture on the resultant viscoelasticity of networks with multiple dynamic bonds is not well understood.” (Page 3)

- *LN 105: 'pendent and telechelic PDMS with a single dynamic bond TYPE were synthesized'*

We have modified the sentence accordingly. (Page 6)

Reviewer 2:

- 1. Currently the work is too physics oriented. It is not easily accessible for those more on the chemistry end of dynamic materials. Diagrams like Figure 1 and Figure 6D are helpful, but it would be stronger if they also existed in between to help the chemistry focused readers better understand the work*

We thank the suggestion from the reviewer. We have adjusted the other Figures to include schematics of the networks in Figure 3 and Figure 4 to improve the clarity. We have also added indicators to the rheology to emphasize the fast and slow processes which are visible from the data.

- The definition of the t_{fast} is not clear. The reviewer can certainly "eyeball" where it should be, but a greater and expanded definition and explanation of it needs to be give. The t_{slow} would be an improvement. It is clearly related to Eq 2, and the generated H function, but again the description is too physics heavy for the broad readership of Nat Comm*

We have taken this opportunity to clarify the meaning of these fast and slow timescales. The fast one is from the maximum in the G'' curve at high frequency, and the slow one is the crossover time. We have modified the text to explain this, and have also added text to the rheology data to point out these processes on Page 9 and 10 of the manuscript.

- Some of the activation energies in Table 2 are extraordinarily small. 10 kJ/mol for imine exchange is very low, and indicates virtually no temperature dependence. Indeed these values seem very low compared to literature values of imine exchange E_a . See Coates et al. 70 kJ/mol (DOI: 10.1039/C9PY01957J). It was only in the presence of very highly withdrawing groups (sulfone) that are quite different from the system studied here which should be much less with drawing. Further analysis and comparison of these results with respect to the literature would be highly beneficial.*

We thank the reviewer for noticing this, and have taken the opportunity to resynthesize and measure the pure imine sample. The new sample shows a more pronounced temperature dependence of the terminal relaxation time, as is shown below.

According to the Arrhenius fit through eq. 1, the activation energy is now 24 kJ/mol. The possible reason of the discrepancy is that the surface of rheological sample last time was a bit rough, which resulted in inaccurate measurement. Additionally, there are issues associated with determining activation energies from relatively small temperature windows and at the limits of the frequency sweep (< 0.01 Hz). We have corrected our temperature dependence of terminal relaxation times in Fig. 4b and the activation energy in Table 2. Although the activation energy is still low, there has been reported literature about such low activation energy of imine bond exchange. For example, Zhou et al. (ACS Appl. Polym. Mater.

2020, 2, 12, 5716–5725) reported lower activation energy of their Vanillin-Based Polyimine Vitrimers, which is 12.3 kJ/mol and 15.96 kJ/mol, respectively. Liang et al. (ACS Sustainable Chem. Eng. 2021, 9, 5673–5683) also reported a low activation energy in their Poly(amide–imine) Vitrimers system. In one of the vitrimers with only imine bond, the activation energy is 20.35 kJ/mol.

We firstly modified Fig. 4c with new data for the pure imine sample and modified its activation energy in Table 2 on Page 14 and Page 15. Also, we have added these references and a discussion of the low activation energies in the imine system on Page 16.

- 4. The overall results are interesting but there appears to be several segments of the literature that are missing. For instance prior work on dynamic materials and architectural effects (E.g. work of Sumerlin, Konkolewicz, and Garcia including reviews) have been completely neglected. Similarly, much of the work in multiply dynamic materials has not been cited (Lehn, Guan etc) including several reviews from the past ~4 years.*

We thank the comment from the reviewer. We have cited several more literature results in the introduction, including from the authors mentioned in the comment:

In terms of the literature (including review) about prior work on dynamic materials and architectural effects, we now cite these articles and review papers:

- 1) Sumerlin et al. (J. Am. Chem. Soc. 2020, 142, 283–289) (Block Copolymer Vitrimers) which emphasize the importance of block architecture on the overall mechanical properties.
- 2) Sumerlin et al. (Polym. Chem., 2018, 9, 2011) (Maximizing the symbiosis of static and dynamic bonds in self-healing boronic ester networks) This paper talks about how permanent crosslinker affects mechanical properties of the polymer dynamic network with boronic ester.
- 3) Konkolewicz et al. (Polym. Chem., 2021, 12, 1975–1982) (ACS Appl. Polym. Mater. 2022, 4, 2, 1475–1486) about how the crosslinking distribution affects the overall mechanical properties.
- 4) Garcia et al. (European Polymer Journal 97 (2017) 120–128) (Effect of the polymer structure on the viscoelastic and interfacial healing behavior of poly(urea-urethane) networks containing aromatic disulphides) about how the effect of polymer structure on the macroscopic viscoelasticity in urea-urethane networks.
- 5) Sumerlin et al. (Progress in Polymer Science 89 (2019) 61–75) (Architecture-transformable polymers: Reshaping the future of stimuli-responsive polymers) (Review paper)

We also gave a literature review about architecture effect in the introduction on Page 3.

In terms of literature about multiply dynamic materials (including reviews), we now cite these articles and review papers:

- 1) Guan et al. (J. Am. Chem. Soc. 2015, 137, 4846–4850) (Enhancing Mechanical Performance of a Covalent Self-Healing Material by Sacrificial Noncovalent Bonds) which introduced a self-healing network self-heal through olefin cross-metathesis and enhance the mechanical property through the sacrificial hydrogen bonding.
- 2) Lehn et al. (Polym Int 2014; 63: 1400–1405) (Double dynamic self-healing polymers: supramolecular and covalent dynamic polymers based on the bis-iminocarbohydrazide motif) which is about the

application of polymers with multiple dynamic bond (dynamic covalent imine bond and hydrogen bond) to make the self-healing material.

3) Konkolewicz et al. (Polym. Chem., 2022, 13, 3705) (Interpenetrated triple network polymers: synergies of three different dynamic bonds) which is about the incorporation of three different dynamic bonds (Diels–Alder, boronic acid-ester and hydrogen bonding) to enhance the mechanical properties.

4) Konkolewicz et al. (ACS Appl. Polym. Mater. 2022, 4, 10, 6850–6862) (Thermo-responsive, Recyclable, Conductive, and Healable Polymer Nanocomposites with Three Distinct Dynamic Bonds) which is about the incorporation of 3 different dynamic bond (hydrogen bonds, Thiol-Michael exchange and Diels–Alder reaction) to fabricate dynamic polymer nanocomposite. The nanocomposite can overcome the trade-off between mechanical stress and elongation.

5) Nicolay et al. (Polymers 2021, 13, 396) (Dually Crosslinked Polymer Networks Incorporating Dynamic Covalent Bonds) (Review paper)

We also added a literature review about multiply dynamic materials in the introduction on Page 4.

5. How effective is water removal? Clearly for imines, water needs to be removed, and varying amounts of water can introduce different exchange mechanisms.

We thank the reviewer for the question about water removal. As mentioned in the synthesis part, the sample was dried in vacuum at 393K for 24 hours. The aim of this procedure is to remove the residue chloroform and water formed during synthesis. After that, the sample was kept in the inert atmosphere to prevent water from getting in. In the Method part, we modify the sentence “The sample was further dried in vacuum at 393K for 24 hours” with the aim of this procedure on Page 27. No water is detectable by our methods including FTIR, NMR, and changes in mass, but there can always be some trace amount even after rigorous drying.

6. Minor issue but sometimes the terminology changes from Pendent to Pendant (Including some figures)

We have corrected all “Pendent” to “Pendant” throughout the manuscript.

Reviewer 3:

Novelty:

1) In the introduction of this work, the authors describe the utility of complex viscoelastic behaviors in a suite of applications (tissue engineering, energy dissipation, adhesion, additive manufacturing, and overall mechanical properties), but the authors do not demonstrate the utility of the reported materials in any of these applications. To attain adequate novelty and impact, a demonstration of the reported materials in use for any of the described applications should be carried out.

To demonstrate that these materials are in fact useful for damping, we have added analysis of the tan delta peaks., as is shown below:

A $\tan \delta \geq 0.3$ is a typical criterion for damping properties of polymeric materials (European Polymer Journal 84 (2016) 770–783). Based on this criterion, six of our samples demonstrate substantial peaks purely from dynamic bond exchange, not T_g as in almost every other damping polymer.

The reported materials also demonstrate improved toughness with mixed dynamic bonds. Tensile tests for two of our samples were conducted by DMA (strain-controlled mode). The pendent networks 100% and 50% imine bonds were tested at a strain rate of 1%/min and are shown below:

The results show that the two samples have nearly the same stress-at-break. However, the mixed network demonstrates larger strain-at-break and larger toughness. The result indicates that by incorporating mixed dynamic bond in the same polymer backbone, the overall mechanical properties can be improved.

To increase the novelty and of the manuscript, we added a new section in the manuscript (Improved mechanical and damping properties in the mixed network) discussing these results on Page 16-18

2) Similarly, as materials are often applied (or at least spend a significant amount of time between activity) under static or pseudo-static conditions at or around room temperature, it may be beneficial to include some general tensile characterization to more easily benchmark these materials.

We agree that the tensile characterization can increase the novelty of the manuscript and have added the above data to the manuscript (Page 17, 18).

3) At several points in the manuscript, the authors note that the key aspects or synthetic design strategies reported in this work have been observed/used in previous reports

- a) *Two relaxation peaks from supramolecular metal-ligand-based materials (Ref 16)*
- b) *Appearance and disappearance of multiple relaxation modes in dynamic materials based on PDMS (Ref 20)*
- c) *Pendant UPy stickers in combination with dynamic covalent moieties (Ref 25)*
- d) *High frequency should relaxation mode in UPy stickers measured via dielectric spectroscopy (Refs 44,45)*
- e) *The most significant overlap comes from reports that have prepared and investigated telechelic PDMS based materials with a mixture of imine and UPy motifs (Ref 28). Granted, this report did not carry out detailed analysis of the rheological properties.*

In its current state, this work serves as an incremental advance in the design of complex viscoelastic materials as the synthesis, design strategy, and overall phenomena have been previously reported. That being said, the authors have put together a very thorough investigation with well-thought out experiments and analysis, it just lacks the novelty required for a broad readership.

The authors wisely state that "... mixing fast and slow bonds does not guarantee well-separated relaxation modes in polymer networks" which sets up an interesting problem in determining design rules for developing such materials. However, this work does not supply sufficient substrate scope or breadth for the rational design of such behavior beyond the prepared materials.

We agree some of these concepts mentioned above appear in the literature. However, these are mostly isolated reports of one system which shows multimodal behavior and there is no broader understanding of what really causes the observed behaviors.

- a) In the work from Grindy et al. (Nature materials 14, 1210-1216 (2015)), multiple relaxation peaks were observed. However, a very similar hydrogel (Advanced Materials 29, 1605947 (2017)) does not show two peaks with boronic ester bonds. The missing knowledge is why two seemingly similar systems do or do not show multiple peaks with mixed dynamic bonds, which is the focus of our manuscript. By comparing identical polymers with the same architecture and comparing with our prior work where fast and slow bonds shared an exchange mechanism, we provide one of the few systematic studies on this problem.
- b) El-Zaatari et al. (Polym. Chem., 2020, 11, 5339) put dynamic bonds with different exchange kinetics into pendant PDMS backbone and claimed that they observed multiple relaxation peaks and provided a conclusion about the condition a when multiple relaxation modes shows up. The only plausible explanation is that a large timescale separation could be causing the observed behavior. However, the $H(\tau)$ in Fig. S12 of this paper is calculated from a stress relaxation curve in Fig. 5a. which has no sign of multiple relaxations and does not fully relax. The stress relaxation data for this mixed network is also noisier and may be leading to the multiple relaxation modes in H. In their conclusion about the multiple relaxation mode, they claimed "If the relaxation times for the individual cross-linkers are within an order of magnitude, the mixed cross-linker system exhibits a single intermediate relaxation mode. When the individual relaxation rates differ by several orders of magnitude, we observe several distinct relaxation modes." In Fig. S11, the individual relaxation rate of MA and BA is separated by more than one

order of magnitude (the separation is even comparable to that from MA and CY in Fig. S12). However, only one relaxation mode shows up in the mixed network with MA and BA.

Even if the multimodal relaxation result is correct in that work, they did not investigate the two key variables in our work: mixed exchange mechanisms and polymer architecture. Our approach to multimodality and understanding design rules is distinct from this prior work.

- c) Although the strategy of incorporation of orthogonal dynamic bonds has been published in Zhang et al. (Macromolecules 53, 5937-5949 (2020)), multiple relaxation modes were not clearly demonstrated. They only looked at one composition of mixed UPy and boronic esters, and did not systematically compare other compositions, architectures, or crosslink densities. So this is again an isolated result without an explanation of why two modes exist.
- d) In this work, we used dielectric spectroscopy to verify the high frequency shoulder measured from rheology come from UPy bond dissociation. Shabbir et al. (Macromolecules 49, 3899-3910 (2016)) and Zhang et al. (Macromolecules 49, 9192-9202 (2016)) just provided the idea of how to verify this. It is a minor point of our manuscript, intended to flesh out the explanation of the UPy relaxation. It is not the reason for our manuscript's novelty, which is the role of exchange mechanisms and polymer architecture.
- e) Although Dai et al. (Advanced Functional Materials 30, 1910723 (2020)) also incorporated UPy and imine crosslinker to the telechelic PDMS, synthesis of UPy motif in that work used different methods so that the final structure of UPy motif in that work (Advanced Functional Materials 30, 1910723 (2020)) is different from ours, and results in the crystallization of UPy group (microphase separation). If such stacking happens, the macroscopic relaxation is not controlled by the UPy dimer dissociation. Instead, it is determined by melting of the UPy stacking. In this case, although the resultant material can have exceptional properties, the polymer dynamics will be difficult to study and they did not probe any multiple relaxation modes. Although the design strategy is similar, the objective of their work is substantially different from ours.

Major:

4) Overall, this manuscript can be difficult to parse. I believe this is, in part, due to a lack of naming scheme for the various materials being studied. I strongly recommend that the authors find a short and legible means of getting across the architecture, amine percentage, and dynamic moiety ratio for easier reading.

We thank the suggestions from the reviewers. We created the new naming system as is shown below:

Pendent networks are named Pend-k-m/n in which k denotes the molar percentage of the amine on the pendant PDMS precursor, and m,n denotes the percentage of amine group functionalized imine crosslinker and UPy motif, respectively. Telechelic networks are named Tele-m/n in which m,n denotes the percentage of amine group functionalized imine crosslinker and UPy motif, respectively.

We described the naming system on Page 6 and Page 7 in the main text. We also modified the naming system throughout the manuscript.

5) As a control, a few materials containing only 50% imine bonds (leaving out the 50% UPy) should be prepared to highlight the lack of faster relaxation modes.

We thank the suggestion from the reviewer. We have synthesized the pendant samples with only 50% of imine and see no fast relaxation mode as shown in the graphs below. Here we see a faster relaxation timescale for the imine bond exchange due to the presence of free amine groups. In the fully crosslinked networks, the bond exchange of imine is through metathesis while in the 50% crosslinked system, it follows the mechanism of transamination (Chem. Sci., 2013, 4, 2253–2261) which is faster.

Based on the reviewer's suggestion, we modified the manuscript, mentioning that with only imine crosslinker and free amine, only one relaxation mode shows up, indicating that the UPy motif is essential to display multiple relaxation modes (Page 10). The graphs shown below are put in SI (Fig S3).

6) Why is the activation energy of the fast mode (as shown in Figure 4b) not analyzed in Table 2 (for the 75/25 and 50/50 samples)? It appears to be similar in magnitude to the highest E_a measured for the imine bond exchange. Is this sensible?

We thank the suggestion from the reviewer. We analyze the activation energy of the fast mode (from UPy) for all the telechelic samples with UPy motif and put all the activation energies with error bars in Table 2 in the manuscript (Page 15 and Page 16). The value of E_a for both bonds is reasonable in the context of prior reports on networks with either an imine or a UPy bond (see response to Reviewer 2, comment 3). The UPy group has a relatively high E_a due to the quadruple hydrogen bonding.

Minor:

7) Figure 1 could use a reaction scheme to show the final chemical structure of the installed UPy groups after reaction with the amine. Also, Figure 1 is quite large and pixelated, and could easily get across the same information in half the space.

We thank the suggestion from the reviewer. We modified Figure 1 with the final chemical structure of the installed UPy groups after reaction with the amine. In addition, we made Figure 1 smaller (Page 7).

8) The legends of certain figures are quite close to data points, which can cause confusion. I suggest surrounding the legends with a box to help separate the legends from data points. A particularly egregious example can be seen in Figure 3c, where the pendant sample legend is effectively collinear with the reported data.

We have surrounded the legends with boxes in all of our figures.

9) The authors went through the trouble of taking an NMR to verify the amine loading, but did not do an independent analysis of the molecular weight of these commercial polymers (which in some cases can deviate significantly from the supplier report).

We have used GPC to measure the molecular weight of the commercially available original PDMS backbone. Since the polymer with amine group can get stuck on GPC column, which increases the elution time, we treated the PDMS backbones with BOC anhydride to protect the NH₂ group before the GPC test. The M_n and PDI are now included in the main text (Page 5, Page 6) and Table 1 (Page 6). Also, the experimental methods have been added (Page 28). The value on the bottle is in good agreement with our measurements. We note that the M_n is not used in any calculations for crosslinking, because it is only the density of NH₂ group in the total sample that are important to determining the stoichiometry.

Reviewers' Comments:

Reviewer #1:

Remarks to the Author:

I read through the authors rebuttal and am supportive of publication.

Reviewer #2:

Remarks to the Author:

Overall the authors have made a strong effort to address the concerns raised by this and other reviewers. The manuscript is much improved.

Reviewer #3:

Remarks to the Author:

The revised manuscript "Polymer Architecture Dictates Multiple Relaxation Processes in Soft Networks with Two Orthogonal Dynamic Bonds" highlights the use of orthogonal dynamic exchange processes (H-bonding and imine) to prepare materials with complex viscoelastic response. While my initial review highlighted concerns regarding novelty, I am happy to report that the authors have more than sufficiently addressed my concerns (as well as the concerns of the other reviewers). Between new experiments and a revised naming scheme, the manuscript now (in my opinion) should appeal to the broad readership of Nature Communications.

I thank the authors for their efforts.